# Desert Soil Microbes as a Mineral Nutrient Acquisition Tool for Chickpea (*Cicer arietinum* L.) Productivity at Different Moisture Regimes

**DOI:** 10.3390/plants9121629

**Published:** 2020-11-24

**Authors:** Azhar Mahmood Aulakh, Ghulam Qadir, Fayyaz Ul Hassan, Rifat Hayat, Tariq Sultan, Motsim Billah, Manzoor Hussain, Naeem Khan

**Affiliations:** 1Department of Agronomy, PMAS Arid Agriculture University, Rawalpindi 46000, Pakistan; qadir@uaar.edu.pk (G.Q.); fayyaz.sahi@uaar.edu.pk (F.U.H.); 2Soil Science Research Institute, PMAS Arid Agriculture University, Rawalpindi 46000, Pakistan; hayat@uaar.edu.pk; 3LRRI, National Agricultural Research Centre, Islamabad 44000, Pakistan; tariqsultannarc@gmail.com; 4Department of Life Sciences, Abasyn University Islamabad Campus, Islamabad 44000, Pakistan; motsimbillah@gmail.com; 5Groundnut Research Station, Attock 43600, Pakistan; manzoorhussainn1199@gmail.com; 6Department of Agronomy, Institute of Food and Agricultural Sciences, University of Florida, Gainesville, FL 32611, USA

**Keywords:** plant growth promoting rhizobacteria, desert, moisture regimes, chickpea

## Abstract

Drought is a major constraint in drylands for crop production. Plant associated microbes can help plants in acquisition of soil nutrients to enhance productivity in stressful conditions. The current study was designed to illuminate the effectiveness of desert rhizobacterial strains on growth and net-return of chickpeas grown in pots by using sandy loam soil of Thal Pakistan desert. A total of 125 rhizobacterial strains were isolated, out of which 72 strains were inoculated with chickpeas in the growth chamber for 75 days to screen most efficient isolates. Amongst all, six bacterial strains (two rhizobia and four plant growth promoting rhizobacterial strains) significantly enhanced nodulation and shoot-root length as compared to other treatments. These promising strains were morphologically and biochemically characterized and identified through 16sRNA sequencing. Then, eight consortia of the identified isolates were formulated to evaluate the growth and development of chickpea at three moisture levels (55%, 75% and 95% of field capacity) in a glass house experiment. The trend for best performing consortia in terms of growth and development of chickpea remained T_2_ at moisture level 1 > T_7_ at moisture level 2 > T_4_ at moisture level 3. The present study indicates the vital role of co-inoculated bacterial strains in growth enhancement of chickpea under low moisture availability. It is concluded from the results that the consortium T_2_ (*Mesorhizobium ciceri* RZ-11 + *Bacillus subtilis* RP-01 + *Bacillus mojavensis* RS-14) can perform best in drought conditions (55% field capacity) and T_4_ (*Mesorhizobium ciceri* RZ-11 + *Enterobacter Cloacae* RP-08 + *Providencia vermicola* RS-15) can be adopted in irrigated areas (95% field capacity) for maximum productivity of chickpea.

## 1. Introduction

In the recent era, climate change had adverse impacts on crop productivity and economic returns from agricultural lands. Water scarcity, drought severity and extreme temperatures have reduced moisture levels in soil profile that alter physiology of the crops resulting in low growth and yield [1]. Thus, for maintaining global food security, it is very important to focus on research to mitigate drought severity. Among the abiotic factors, drought is generally thought to be a leading factor for decline in growth of crops [2]. The world population is increasing day by day and is likely to further increase more than 9 billion by 2050, and food availability is imperative, thus the need to focus on minimizing the limiting factors of crop productivity in limited arable land resources [3,4,5]. To date, more attention is given to soil microbe association with plants during prolonged spells of abiotic stress for enhanced net return of field crops [6]. Studies of the interactions between plants and their microbiome have been conducted worldwide in the search for growth-promoting representative strains to be used for biological inputs for agriculture, aiming to achieve more sustainable agriculture practices [7]. Among the most important food pulses, chickpea (*Cicer arietinum* L.) is a major source of protein which is an essential part of human diet [8]. The total area under chickpea cultivation in the world is 17.81 million hectares giving a very less production (17.19 million tons) of chickpea [9]. The decreased economic return of chickpea may be attributed to moisture deficiency and nutrient deficient soils [10].

Use of efficient bacteria in the leguminous crops for acquisition of nutrients is a way forward to enhance crop yield and drought stress tolerance in plants. Among leguminous crops, the chickpea plants fix about 60% of available nitrogen from the environment through the process of biological nitrogen fixation [11]. *Rhizobium* strains increased crop growth rate significantly when used in combination with phosphorus solubilizing bacteria as compared to their sole strain functions [12]. Phosphorus comes after nitrogen for nutritional requirements of plants and is one of the most important minerals required for plant growth occupying a strong position among soil macronutrients [13]. Phosphorus plays a key role in root growth and development, stimulating cell division or cell elongation. Due to phosphorus availability, the surface area, volume and root length increases, thus phosphorus indirectly promotes water absorption from untapped soil profile. However, in drought conditions, P-accumulation or fixation occurs due to phosphatase enzyme inactivity [14]. Phosphate solubilizing microbes release phosphatase enzyme, which is responsible for P-solubilization from fixed soil P-pools for plant P acquisition [15,16]. Thus, phosphate solubilizing microbes are a major solution for mineralization of soil fixed P in drought conditions [17]. Moreover, about 80% of applied phosphorus in fertilizers gets fixed in alkaline calcareous soils of Pakistan. However, soil P sorption and desorption capacity, particle size distribution, metal oxides, and soil pH are also main factors regulating P dynamics in the soil [18]. Generally, soils hold enough quantity of phosphorus but in unavailable form [19]. Thus, PGPR (Plant Growth Promoting Rhizobacteria) having P-solubilizing activity, mineralizing insoluble phosphorus into soluble form by producing organic acids, contribute to a massive increase in the crop yield [20]. Considering the key role of phosphorus in root growth and development [16], an increase in its availability and consequent potential extensive root growth of plants will be helpful for water absorption from untapped soil profiles, an important goal for high yield of chickpea crop in Thal Desert, Pakistan, where drought stress is a major obstacle.

It has been reported that PGPR have significant positive effects in alleviating drought stress in dry land crops by producing 1-aminocyclopropane-1-caboxylate (ACC) deaminase, exopolysaccharides (EPS) and phytohormones, contributing in the acquisition of drought tolerance in plants [21]. Thal desert is located (31°30′00.0″ N 71°40′00.0″ E) in Tehsil Nurpur Thal, Mankaira, Bhakkar and Chobara along with some part of districts Jhang and Muzaffargarh, Punjab, Pakistan. The chickpea producing area “Thal” falls under rain-fed areas, and crop production depends upon annual rainfall as well as on some patches of the area under tube well irrigation system. The average production of chickpeas under a rainfed environment is not so impressive. Leguminous crops have specific symbiotic systems which are capable to exist in extreme conditions of alkalinity, temperature and moisture stress [10] which shows its adaptability to different soil and environmental conditions. Atmospheric nitrogen fixation through symbiotic association in the root nodules of leguminous crops is the major substitute of inorganic nitrogenous fertilizers in the vast tract of Thal desert, Pakistan. However, considering the scarcity of water and issues of reduced crop yield from the deserts, the present study was planned to estimate the role of potential bacteria to enhance the grain yields of chickpea in Thal Desert under drought conditions. The hypothesis of our study is that isolated bacteria may be helpful to increase the drought tolerance in plants by secreting different compounds like abscisic Acid (ABA), exopolysaccharides, producing ACC-deaminase and enzymatic activities. Moreover, *rhizobium* species may increase nodulation for better productivity in chickpeas through the process of biological nitrogen fixation. Building consortia between PGPR, including those having P-solubilizing activity and rhizobium species might increase significantly chickpea crop production.

## 2. Results

### 2.1. Morphological Characterization of Isolated Strains

Initially, 72 strains were tested for screening, and among them, the 6 most promising strains were observed by Phase Contrast Microscope (Phase contrast 2, Nikon, Japan) for the colony morphology. MIRA3, Tescan Libušina třída, Brno, Czech Republic SEM (scanning electron microscope) was used to characterize colony shape, form, elevation, color and margin (Table 1).

### 2.2. Biochemical Characterization of Isolated Strains

Identified bacterial strains were biochemically characterized. Two strains (RZ-11 and RZ-22) showed positive results for ammonia production whereas others remained negative for this test. The appearance of yellowish colored zones around the colony indicated siderophore production. Isolates RP-01, RP-08, RS-14 and RS-15 were found positive for siderophore tests while the other two (RZ-11 and RZ-22) appeared negative. Regarding the hydrogen cyanide (HCN) test, all strains tested negative. The appearance of cherry red rings in the tube indicated positive signs for IAA production. Among the isolates, 2 strains (RP-08 and RS-15) were found strong positive, while RP-01 and RS-14 were observed as moderate IAA producers and *Rhizobium* isolates tested negative. However, for solubilization of tri-calcium phosphate, 2 strains (RP-01, RP-08) were found strongly positive with solubilization Intensity (200 and 190) and solubilization index % (3.00 and 2.90) whereas, RS-14 and RS-15 did not show P-solubilization activity. Moreover, 4 PGPR strains (RS-14, RS-15, RP-01, RP-08) revealed strong potential for the catalase and protease enzymes production along with ACC-deaminase and EPS activities. Similarly, 3 strains RP-01, RS-14 and RS-15 showed positive response for amylase production. (Table 2).

### 2.3. Molecular Identification of Isolates

Most efficient rhizobium (RZ-11 and RZ-22) and PGPRs (RP-01 and RP-08, RS-14 and RS-15) strains were identified using 16sRNA gene sequencing analysis. The identified nucleotides sequences were arranged in the phylogenetic tree (Figure 1) by using MEGA 7 [22]. Isolate RZ-11 matches to *Mesorhizobium ciceri* strains RZ-22, SS1 (5) and CM-25. Similarly, strain RS-14 has matching criteria to bacteria *Bacillus mojavensis* strain RS-1, PMCC-9 and LMB3G43 in the phylogenetic tree and RS-15 matches 100% to *Providencia vermicola* and is clustered together with 3 *Providencia* strains Mum1, Ag1 and OF6. PGPR isolates having P-solubilizing ability coded RP-01 and RP-08 match to *Bacillus subtilis* strain XGL205 and *Enterobacter cloacae* strain MSK, respectively.

### 2.4. Quantification Assay of ACC-Deaminase, EPS, IAA and P Solubilization

Most promising PGPR strains (RP-01, RP-08, RS-14 and RS-15) produced ACC-deaminase, Exopolysaccharides (EPS) and Indole acetic acid (IAA) very effectively (Table 3). However, the strain RP-08 showed maximum values of ACCD, EPS and IAA, which averaged 0.84 (μM/mg protein/h), 0.80 (mg/mL) and 86 (μg/mL), respectively. Table 3 showed that T_2_ consortium was found efficient for ACCD and IAA (2.6 μM/mg protein/h and 177 μg/mL), respectively, followed by T_7_ with ACCD (2.5 μM/mg protein/h) and IAA (171 μg/mL). Isolate RP-01 showed the highest value (14.2 ug/L) of P-solubilization compared to other strains (Figure 2).

### 2.5. Growth Chamber Experiment for Screening of Isolates

Seventy-two isolated strains were evaluated in a trial conducted in growth chambers to screen out most efficient rhizobium and PGPR strains for different attributes of seedling growth (Figure 3). Two isolates (RZ-11 and RZ-22) significantly affected the nodulation of chickpeas that produced 10.667 and 10.567 nodules per plant, respectively (Figure 4). Among all tested isolates, two PSB strains (RP-01 and RP-08) significantly enhanced shoot length by 19.4 cm and 19.5 cm, respectively (Figure 5). A similar trend was recorded in terms of root length (15.10 cm and 15.133 cm) by the same bacterial strains, respectively (Figure 6). Two PGPR isolates (RS-14 and RS-15) were found most efficient for growth and development of the plant shoot and root in screening trials. Figure 7 shows maximum shoot length (22.067 cm) in the treatment where seeds were inoculated with RS-14 followed by RS-15 (21.6 cm). On the other hand, the highest root length (19.033 cm) was recorded for the treatment where chickpea seeds were inoculated with RS-15 followed by RS-14 (17.667 cm) (Figure 8).

### 2.6. Glass House Experiment at Different Moisture Regimes

The isolates to be used for making each consortium appeared compatible with each other, thus consortia were built and inoculated to elucidate prominent impact on nodulation, growth and yield attributes of chickpea at different moisture regimes in pots. Maximum number of nodules plant^−1^ (12.33), plant height (32.37 cm), number of pods pot^−1^ (20.67), root length (29.61 cm), 100 grain weight (29.84 g), biological yield (38.93 g pot^−1^), grain yield (17.90 g pot^−1^) and harvest index (44.67%) were recorded with T2 (*Mesorhizobium ciceri* RZ-11 + *Bacillus subtilis* RP-01 + *Bacillus mojavensis* RS-14) at moisture level 1 (55% of soil field capacity) (Table 4). However, grain yield obtained under this treatment was at par to the un-treated control (T9) at moisture level 3 (95% of field capacity). Similarly, statistically higher biological and grain yield (23.17 g pot^−1^) was obtained by T7 (*Mesorhizobium ciceri* RZ-22 + *Enterobacter cloacae* RP-08 + *Providencia vermicola* RS-15) at moisture level 2 (75% of soil field capacity) compared to control (17.10 g pot^−1^) followed by T_4_ (*Mesorhizobium ciceri* RZ-11 + *Enterobacter Cloacae* RP-08 + *Providencia vermicola* RS-15) at 95% of field capacity.

#### 2.6.1. Physiological Attributes and Nutrient Acquisition of Chickpea

The data pertaining to proline content in leaves of chickpea (Table 5) show that the treatment T2 gave maximum proline content (4.7967 mg g^−1^ DW) at moisture level 1 (55% FC) followed by T7 (4.6067 mg g^−1^ DW) at same moisture level, while minimum proline contents (1.1533 mg g^−1^ DW) were determined from untreated plants at moisture level 3 (95% FC). Maximum grain N and protein contents (4.31% and 26.96%) were recorded for T_4_ at moisture level 3 (95% FC) followed by 4.28% and 26.79% with application of T_7_ at moisture level 2 (75% FC), respectively (Table 5). T_2_ responded most effectively at moisture level 1 (55% FC) both for grain N and protein (4.19% and 26.17%) contents. Lower N and protein contents were observed in grains of untreated plants at all given moisture levels. Similarly, T_7_ was the most promising consortium on moisture level 2, showing 0.35% P contents in grain, followed by T_4_ at moisture level 3. T_2_ was the best performing consortium among all treatments, including the control at moisture level 1 which resulted in 0.31% P in grains of chickpea. The data regarding nitrogen acquisition in chickpea shoot shows that the maximum nitrogen contents (1.66–1.65%) were recorded for treatments T_2_ and T_4_ at moisture level 1 (55% of field capacity). Similarly, T_7_ appeared as the most efficient consortium with 1.65% N content in chickpea shoot at moisture level 2. Contrarily, T_9_ (control) showed lowest N contents (1.31–1.32%) in shoot of chickpea at all moisture levels. The data pertaining to the P contents in chickpea shoot (Table 5) showed the highest P (0.34%) for T_4_ at moisture level 3 followed by T_2_ which showed 0.32% P at the same moisture level. Similarly, T_7_ gave 0.30% P contents at moisture level 2. T_9_ (control) was recorded as the lowest performing treatment, which showed 0.20–0.22% P contents in chickpea shoots at all given moisture regimes.

#### 2.6.2. Isolates Survival in Rhizospheric Soil

The data regarding the isolate population in rhizospheric soil (Figure 9) shows that the maximum Colony Forming Unit (2.45 × 10^8^) value was recorded for the treatment T_2_ at moisture level 3 followed by T_7_ with 2.36 × 10^8^ at the same moisture level. T_2_ and T_7_ showed 1.67 × 10^8^ and 1.36 × 10^8^, respectively, at moisture level 1. T_9_ (control) showed zero population of the tested consortia isolates at all given moisture levels.

#### 2.6.3. Post-Harvest Soil Nutrient Status

The data regarding the nitrogen percentage in post-harvest soil (Figure 10) shows that the highest N (0.016%) was recorded for the treatments T_4_, T_5_ and T_7_ at moisture level 3 followed by T_2_ with 0.0157% N in post-harvest soil at moisture level 1. Similarly, T_7_ gave 0.0157% N at moisture level 1 and 2. T_9_ (control) was recorded as the lowest performing treatment, which showed 0.012% N in post-harvest soil, at all given moisture levels. The data pertaining to phosphorus percentage in post-harvest soil (Figure 11) shows that the maximum phosphorus (4.5 ppm) was recorded for both treatments T_2_ and T_7_ at moisture level 1 (55% of field capacity). T_7_ was recorded as the most efficient consortium at moisture level 3 (4.166 ppm post-harvest soil P). T_9_ (control) showed the lowest P contents (3.00 ppm) in post-harvest soil at all moisture levels.

## 3. Discussion

In this study, we isolated 125 bacterial strains from collected samples of nodules, rhizoplane and rhizospheric soil of chickpea from the Thal desert of Punjab, Pakistan during a severe drought spell. Out of all isolated strains, 72 were selected for screening. Most efficient strains were biochemically tested, and RZ-11 and RZ-22 (*Mesorhizobium ciceri*) produced ammonia. Similar findings were given by [23] who showed that some soil microbes produce ammonia and enhanced crop growth attributes. Two PGPR isolates RS-14 (*Bacillus mojavensis*) and RS-15 (*Providencia vermicola*) were evaluated as IAA producing bacterial strains. Several microbes produce active auxin as IAA, which is a plant growth promoter [24,25]. Two PGPR isolates RP-01 (*Bacillus subtilis*) and RP-08 (*Enterobacter cloacae*) were IAA producers and the most efficient phosphate solubilizers. The results are also in agreement with the findings of [26], who observed the maximum phosphate solubilizing ability for *Enterobacter* sp. The appearance of hallo zones surrounding the microbial colonies could be due to the synthesis of organic acid with low molecular weight, or due to polysaccharides and phosphatase production by phosphate solubilizing microbes [27]. In our study, most promising strains (RZ-11, RZ-22, RP-01, RP-08, RS-14, RS-15) produced exopolysaccharides (EPS), phytohormones, 1-aminocyclopropane-1-1 carboxilate (ACC) deaminase and helped the chickpea in acquisition of drought tolerance at 55% field capacity. Similar studies were conducted by [28,29] on the mechanism to induce drought tolerance in wheat and chickpea grown on dry lands. They highlighted the role of microorganisms to manage abiotic and biotic stress by producing indole acetic acid (IAA) and ACC-deaminase to reduce ethylene levels of in roots.

In the growth chamber assay for isolates’ screening, nodulation plant^−1^ was increased over 100% by rhizobium strains (RZ-11, RZ-22) as compared to un-inoculated plants of chickpea grown in pre-autoclaved soil. Similar results were found by [30,31] who reported that *Mesorhizobium ciceri* inoculation increased significantly the nodulation of chickpea plants through symbiotic relationship. These results are in conformity with the results of [32] who recorded increase in chickpea nodulation by seed inoculation with *Mesorhizobium ciceri*. PGPRs having P-solubilizing activity (RP-01 and RP-08) showed significant enhancement in chickpea shoot and root length by 54.4% and 54.9% over the un-inoculated control. Similar findings on phosphate solubilizing bacteria were given by [33], who noted that PSB enhances shoot and root length of chickpea plants. The root and shoot length may be increased due to increased availability of phosphorus thanks to PSBs activity, as phosphorus has an important role in root development and cell division. All plant growth promoting isolates showed significant improvement in seedling shoot length, between 6.39% and 45.82%, as compared to the control; two PGPR isolates, RS-14 and RS-15, having phytohormonal activity, increased chickpea shoot length 42.73% and 45.82%, respectively, in comparison to the untreated control. Moreover, PGPR strains (RS-01 to RS-24) showed a significantly positive response in chickpea seedlings root length (from 0.55% to 56.87%) as compared to the control. However, the maximum root length of chickpea was statistically increased 58.6% and 47% by RS-15 and RS-14, respectively, over the control without inoculation. These results are in conformity with the findings of [34] who reported that phytohormone production is the main character of shoot-root increasing strains in chickpea plants. Our experimental results are also supported by findings of Marasco et al. [35], who showed that plant-root system increases up to 40% higher in PGPRs treated plants as compared to untreated controls.

Six most efficient strains were used to make 8 different prolific consortia that were evaluated in pot experiment to study their efficiency on growth and yield attributes of chickpea variety Bhkkar-2011 at 3 moisture regimes (55%, 75% and 95% of field capacity). T_2_ consortium (*Mesorhizobium ciceri* RZ-11 + *Bacillus subtilis* RP-01 + *Bacillus mojavensis* RS-14) was found most effective for chickpea growth and yield attributes at moisture level 1, which was maintained on 55% of field capacity. The treatment showed 44% increase in plant height, 29% in number of pods pot^−1^, 19.8% in root length, 29% in 100 grain weight, 46.9% in biological yield pot^−1^, 50% in economic yield pot^−1^ compared to the untreated control at moisture level 1 (55% of FC). Increments in the yield and yield attributes of chickpea might be due to ammonia production by rhizobium, and IAA production, ACC deaminase and PSB activity of PGPRs from the consortia in the applied treatment. Our results are supported by [36] who took samples of Rhizospheric soil and root nodules from a selected chickpea field and revealed that grain yield and yield attributes were increased by inoculation with Rhizobium strains in comparison with the un-inoculated treatment. The physiological attributes of chickpea as grain N and protein content were statistically increased by 7.6% over the control, with application of T_2_ at moisture level 1 (55% FC) and by 10.12% (grain N) with T_4_ at moisture level 3 (95% FC). T_2_ resulted in postharvest soil N and P increases of 30% and 50%, respectively, in comparison with the un-inoculated treatment at 55% of field capacity. Moreover, in the present study, the calculated proline content of chickpea leaves showed an increase at moisture level 1 (55% of FC) by inoculation with T_2_-consortium having ACCD activity, and IAA production. These results are in accordance with findings of Mandhurendra [37]. Statistically higher grain yield (35.49%) over that of untreated chickpea plants was obtained with T_7_ treatment (*Mesorhizobium ciceri* RZ-22 + *Enterobacter Cloacae* RP-08 + *Providencia vermicola* RS-15) at moisture level 2 (75% of FC) and with T_2_ (*Mesorhizobium ciceri* RZ-11 + *Bacillus subtilis* RP-01 + *Bacillus mojavensis* RS-14) (50% higher grain yield) at moisture level 1 (55% of FC). T_4_ (*Mesorhizobium ciceri* RZ-11 + *Enterobacter Cloacae* RP-08 + *Providencia vermicola* RS-15) had 27.53% better economic yield at moisture level 3 (95% of FC) as compared to uninoculated chickpea plants. Hence, microbial combinations inT_2_ and T_7_ could be used to make effective biofertilizers for chickpea growing areas under rainfed conditions to help plants to cope with drought spells.

## 4. Material and Methods

### 4.1. Study Area and Sample Collection

Sample collection was done from different major chickpea producing tracts of Thal desert, Pakistan during cropping season 2017–2018. Roots of five healthy plants having nodules, rhizospheric soil (RS) and rhizoplane soils (RP) were uprooted from sand dunes of each location during drought spell (soil moisture ≈ 7%).

### 4.2. Isolation of Rhizobium and PGPRs

The rhizobium strains were isolated from pink colored nodules of collected roots of chickpea plants. A milky suspension was obtained after dissolving the crushed nodules in 5 mL distilled water. A droplet of milky suspension (100 μL) was shifted to Yeast Mannitol Agar (YMA) plates [38], and colonies were grown on plates after incubation at 28 ± 2 °C for 7 days. PGPR were isolated from rhizoplane (RP) and rhizospheric soil (RS) of collected roots by using serial dilution technique on Pikovskaya agar and Luria Bertani medium [39], respectively. The colonies were incubated at 28 ± 2 °C for 24 to 48 h.

A total of 125 rhizobacterial strains were isolated from root nodules (RZ), rhizospheric (RS) and rhizoplane (RP) soil of chickpea plants. Among them, 72 were repeatedly streaked on respective mediums in petri plates to obtain pure colonies for a screening process in growth chambers.

### 4.3. Morpho-Physiological Characterization of Isolated Strains

Colony morphologies of Rhizobium and PGPR isolates were observed by spreading pure colonies on respective media for their shape, margin and color [40]. Further strains were characterized as gram positive or gram negative via gram staining protocol [41].

### 4.4. Biochemical Characterization of Isolates

Bacterial strains were cultured in Luria Bertani (LB) broth with tryptophan to test the Indole Acetic Acid (IAA) production that was confirmed by adding 5 drops of Kovac’s reagent directly to the respective tubes containing bacterial isolates [42]. However, isolates were evaluated for ammonia production by the method of Dinesh [43]. Bacterial strains were also screened for siderophore and hydrogen cyanide (HCN) production by using the methods adopted by [44] and [45], respectively. Phosphate solubilization efficiency and index of isolates were determined by the method of Macfaddin [46].


(1)$$PSE=\frac{Colony\;diameter+\mathrm{Halozone}\;\mathrm{diameter}}{Colony\;diameter}$$
(2)$$PSI=\frac{Diameter\;of\;clearance\;zone}{Diameter\;of\;growth\;zone}\times 100$$


Bacterial strains were tested for catalase production using the procedure given by [47]. For this purpose, a loop having fresh bacterial strains was placed on glass slides. Hydrogen peroxide (H_2_O_2_) was added on each spot of isolates, and production of bubbles was counted as a positive sign of catalase enzyme. For the amylase test, iodine solution was added on developed bacterial colonies to observe the formation of clear halo zones that indicated amylase production [48]. Skimmed milk agar medium (SKM) was used to observe the protease production for isolates by adopting the method described in [49]. For determination of ACC deaminase enzyme activity, bacterial strains were grown in Tryptic soy broth (TSB) for 24 h and centrifuged at 3000 g. Then, a loop of the bacterial strains suspended in Tryptic Soy Broth (TSB) media was shifted onto sterile DF (Dworkin and Foster) salt media containing ACCs as a single source of nitrogen Afterwards, the plates with the bacterial strains were incubated for 3 days at 28 °C and checked for colony growth [50]. ACC-deaminase activity was quantified as nmol α-ketobutyrate mg protein^−1^ h^−1^ using a spectrophotometer at 540 nm wavelength [51]. Exopolysaccharide (EPS) producing bacterial strains were characterized by streaking them on American Type Culture Collection (ATCC) medium no.14 and incubated for 3 days at 28 °C. Consequently, the bacterial colonies showing slimes around them were characterized as EPS producing strains [52].

### 4.5. Growth Chamber Experiment for Screening of Isolates

A plastic jar experiment in a growth chamber was carried out in the laboratory of Land Resources Research Institute (LRRI), National Agriculture Research Centre, Islamabad to evaluate the effect of isolates, 24 Rhizobium (RZ-01 to RZ-24) and 48 Plant Growth Promoting Rhizobacteria) on growth attributes of chickpea seedling. The chosen PGPRs were isolated from rhizoplane soil (RP-01 to RP-24) and from rhizospheric soil (RS-01 to RS-24) to screen out 2 best strains from each category. The jars were sterilized with 20% sodium hypochlorite solution and filled with pre-autoclaved sandy soil (at 120 °C for 90 min). In each jar, three surface sterilized seeds of variety Bhakker-2011 inoculated with respective isolates were sown. Uninoculated seeds were sown in the jar which was designated as control treatment. These jars were kept in the growth chambers in a complete randomized design. The data regarding nodulation per plant, shoot length and root length of all treated plants were recorded. Among the isolates, two best Rhizobium and four PGPRs were chosen on the basis of their effectiveness on growth attributes of chickpea. The chosen isolates were molecularly characterized and identified for further experimentation to evaluate their consortium effects on the productivity of chickpeas at different moisture regimes.

### 4.6. Molecular Characterization of Selected Strains

The most efficient strains of rhizobium (RZ-11, RZ-22) and PGPRs (RP-01, RP-08, RS-14, RS-15) from the previous screening experiment were molecularly identified by amplification and sequencing of 16sRNA gene. Purified colonies of the most efficient bacterial strains were plucked and mixed with 20 µL Tris-EDTA buffers in Polymerase Chain Reaction (PCR) strips. This mixture was placed in a PCR apparatus (Thermal Cycler PCR PEQSTAR, Munich, Germany) for 10 min at 95 °C to extract the template DNA, which was collected in the supernatant after centrifugation. DNA amplification was carried in the same apparatus using 2 µL of forward and reverse universal primers as 9F (5′-GAGTTGATCCTGGCTCAG-3′) and 1510R (5′-GGCTACCTTGTTACGA-3′), respectively, 25 µL TAKARA Pre-mix Ex-Taq, 20 µL PCR water and 1 µL of DNA template. The amplified PCR products were sent to Macrogen, Seoul Korea, for sequencing, and strains were identified using the EzBioCloud server Macrogen, Seoul Korea. All sequences were submitted to gene bank for allotment of accession numbers.

### 4.7. Compatibility of Isolates for Consortia

Three bacterial strains in consortium were grown jointly on nutrient agar medium (3.0 g yeast extract, 5.0 g peptone and 20.0 g agar L^−1^) to test their compatibility. For that purpose, isolates were refreshed overnight on 25 mL of Nutrient broth (NB) medium and inocula of 2 µL, containing 10^6^ bacterial cells of a distinct isolate were inoculated 1 cm apart from other isolate on nutrient agar (NA) medium in one petri plate. Such petri plates in triplicate were incubated at 30 °C for 72 h. The inhibiting effect of isolates was examined visually using culture images and close ups of the overlapping areas of expanding colonies as adopted by [53].

Moreover, ACC-deaminase activity, IAA production and phosphate solubilization activity of consortia were quantified using the methods as discussed in Section 2.4. Glass house experiment at different moisture regimes.

A pot experiment was conducted under control conditions at Plant Genomic Research Institute (PGRI), NARC Islamabad to evaluate the efficacy of characterized Rhizobium and PGPRs on growth and yield attributes of chickpea in comparison to the untreated control. For the experiment, the sandy loam soil was collected from the Thal desert and passed through a sieve of 2 mm diameter. The pots with equal sizes were filled with 8 kg autoclaved soil. Three moisture regimes (55%, 75% and 95% of field capacity) were calculated by determining the porosity, bulk density and volume of pots with the help of formula in [54,55], and the moisture was maintained with the help of Time Domain Reflectometer (TDR) used during the experiment.

Three surface sterilized seeds of Bhakker-2011 (both inoculated and uninoculated) were sown in respective pots, and the experiment was followed with all the three germinated seeds. Experiment with three replicates of each treatment was conducted in CRD (Completely Randomized Design) under controlled conditions. Pure cultures of rhizobium strains and Plant Growth Promoting Rhizobacteria (PGPR) and the PGPRs having phosphate solubilizing activity were inoculated individually in 50 mL NB and further incubated overnight at 30 °C, 150 rpm. The 24 h old cultures of each isolate were shifted into a sterile 50 mL tube and centrifuged at 2150 g for 3 min. The resulting pellets were washed 5 times with sterile distilled water [53]. The resulting washed pellets of each isolate were resuspended in sterile water, and the optical density was measured at 550 nm and adjusted to a concentration of 10^6^ CFU mL^−1^. Strain suspensions were combined in equal amounts (i.e., equal cfu mL^−1^) to prepare the respective consortium for each treatment. Bacterial strain consortia (Table 6) were tested for their efficacy to improve chickpea productivity with following set of treatments:

### 4.8. Seed Inoculation with Consortia

The chickpea seeds were washed with sterile water and rinsed with 70% ethanol. Seeds were then immersed in 6.5% sodium hypochlorite and agitated for 20 min. Afterwards, seeds were washed eight times with sterile water under sterile conditions. The washed seeds were soaked in respective consortia for 60 min. The seeds for the untreated control were soaked in sterile water (Molina Romero et al.) [56].

#### 4.8.1. Isolate Survival in Rhizospheric Soil

Survival of the inoculated insolates was determined from rhizospheric soil of the chickpea plants 60 days after seed inoculation through plate count method. Calculations were performed on the basis of serial 10 folds dilution in duplicate, using the pour plate method, replicated samples of 1 g soil and an appropriate dilution [57]. Each value is presented as an average of three individual plate counts of the colonies of the PGPR isolates within a consortium. Petri dishes (90 mm diameter) contained 25 mL of Nutrient Agar (NA) medium, and plates were incubated at 28–30 °C. Colony forming units (CFU) were recorded after 48 h, and the average number per g oven dry weight of soil was calculated as
(3)$$\mathrm{CFU}=\frac{Bacterial\;plate\;count\;\times \;\mathrm{Dilution}\;\mathrm{factor}}{Oven\;dry\;weight\;of\;soil}$$

#### 4.8.2. Soil Properties

Three composite samples were collected from the pile of pre-sowing soil for physico-chemical analysis; soil texture [58], organic matter [59] total phosphorus [60], total nitrogen [61], extractable phosphorus and extractable potassium [62], and soil pH (1:5 soil-water) were determined following the methodology described by [63]. The mean of the initial soil replicated data is mentioned in Table 7. The post-harvest soil properties (Total Nitrogen and Extractable phosphorus) were determined by adopting the methods described in [61,62], respectively.

#### 4.8.3. Leaves Proline and Grain Protein Contents

Proline contents in 130 days old leaves were determined following the method described in [64]. Samples were weighed, and proline was extracted in sulphosalicylic acid and evaluated using the ninhydrin reagent. Two layers were obtained in separating funnels during the estimation process, and the upper pink layer was selected for quantification using a spectrophotometer at 520 nm wavelength. Seed protein contents were analyzed following the method described in [61].

### 4.9. Statistical Analysis

The data regarding biological and economic yield attributes of chickpea grown in jars and pots under growth room chambers, and glass house conditions were analyzed statistically by adopting a Complete Randomized Design (CRD). The means were separated with Least Significance Difference (LSD), and the comparison of the treatment means was done through LSD test [65].

## 5. Conclusions

Extreme drought events are expected to be one of the main challenges for agriculture and a threat to global food security. Explo/ration and utilization of desert microbes to cope with the issue of drought through experimentation on desert soil is a valid idea being adopted in the present study. Here, a series of experiments revealed growth promotion as well as substantial nodulation in chickpea that enhanced its grain yield under drought stress. This approach indicates the vital role of isolated strains to be utilized as bio-fertilizers under drought spell in the Thal desert, a main chickpea producing area in Pakistan, which is the 4th largest chickpea producing country. In this study, we found that the consortium T_2_ (*Mesorhizobium ciceri* RZ-11 + *Bacillus subtilis* RP-01 + *Bacillus mojavensis* RS-14) can perform best in drought conditions (55% field capacity), and T_4_ (*Mesorhizobium ciceri* RZ-11 + *Enteroabacter Cloacae* RP-08 + *Providencia vermicola* RS-15) can be adopted in irrigated areas (95% field capacity) for maximum productivity of chickpea.

## Figures and Tables

**Figure 1 plants-09-01629-f001:**
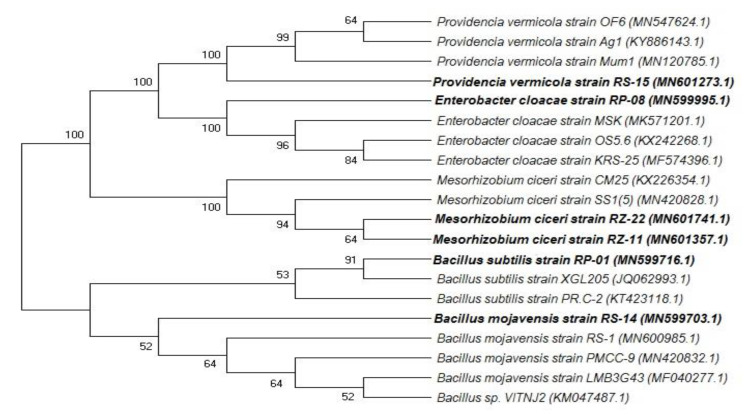
Phylogenetic tree showing the screened isolates given in bold letters and closest homologues. Bootstrap values (*n* = 100) are displayed at the nodes.

**Figure 2 plants-09-01629-f002:**
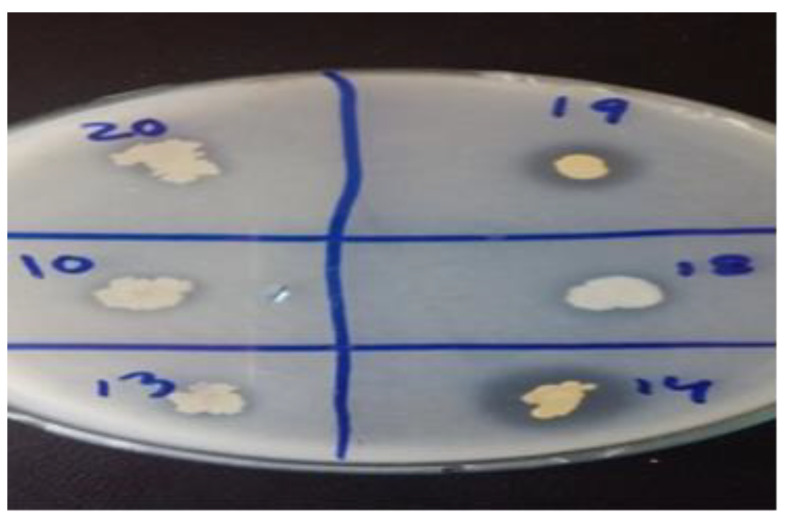
Efficient strains showing halozones for phosphate solubilization, Halozone 14 = RP08 (*Enterobacter cloacae*), Halozone19 = RP01 (*Bacillus subtilis*).

**Figure 3 plants-09-01629-f003:**
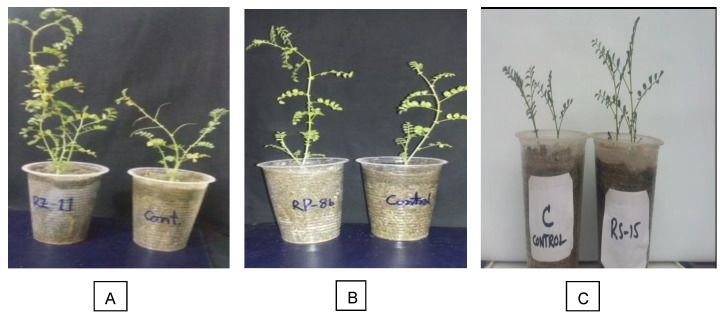
Effect of strains on chickpea shoot length in growth chamber, where (**A**) *Mesorhizobium ciceri*-RZ11, (**B**) *Enterobacter cloacae*-RP08, (**C**) *Providencia vermicola*-RS15.

**Figure 4 plants-09-01629-f004:**
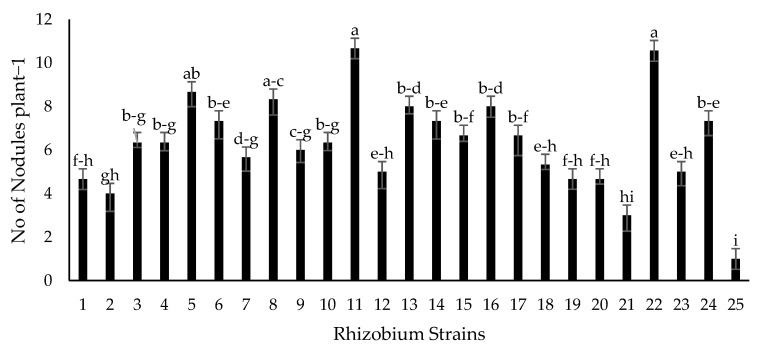
Effect of Rhizobium strains on nodulation plant^−1^. Strains (1–24) are rhizobium (RZ-01 to RZ-24) while 25 is the control treatment. The bars indicate standard error (±SE) of the mean (*n* = 3). Means followed by the same letter within the bars are not significantly different at *p* = 0.05.

**Figure 5 plants-09-01629-f005:**
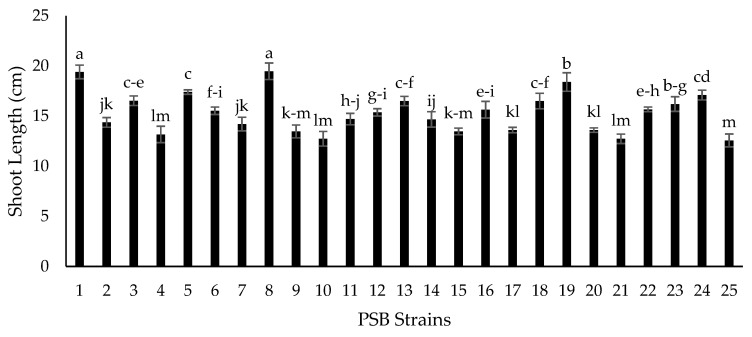
Effect of phosphate solubilizing bacteria (PSB) strains on chickpea shoot length (cm). Strains (1–24) are phosphate solubilizers (RP-01 to RP-24) while 25 is the control treatment. The bars indicate standard error (±SE) of the mean (*n* = 3). Means followed by the same letter within the bars are not significantly different at *p* = 0.05

**Figure 6 plants-09-01629-f006:**
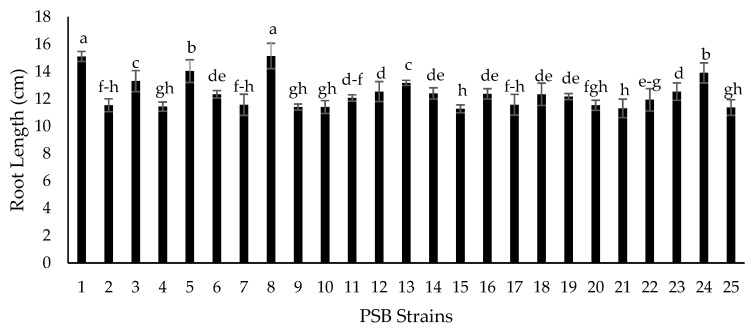
Effect of PSB strains on chickpea root length (cm). Strains (1–24) are phosphate solubilizers (RP-01 to RP-24) while 25 is the control treatment. The bars indicate standard error (±SE) of the mean (*n* = 3). Means followed by the same letter within the bars are not significantly different at *p* = 0.05.

**Figure 7 plants-09-01629-f007:**
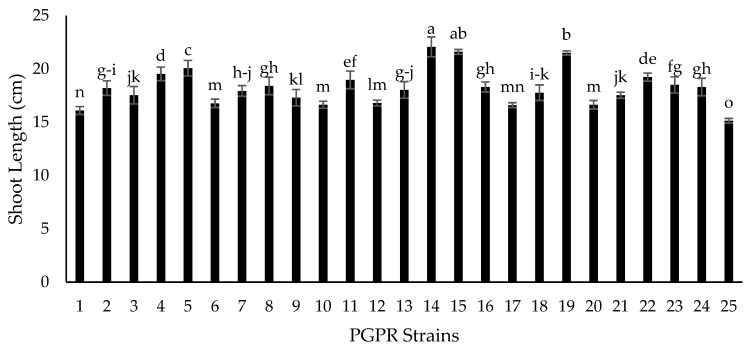
Effect of plant growth promoting rhizobacteria (PGPR) strains on chickpea shoot length (cm). Strains (1–24) are PGPRs (RS-01 to RS-24) while 25 is the control treatment. The bars indicate standard error (±SE) of the mean (*n* = 3). Means followed by the same letter within the bars are not significantly different at *p* = 0.05.

**Figure 8 plants-09-01629-f008:**
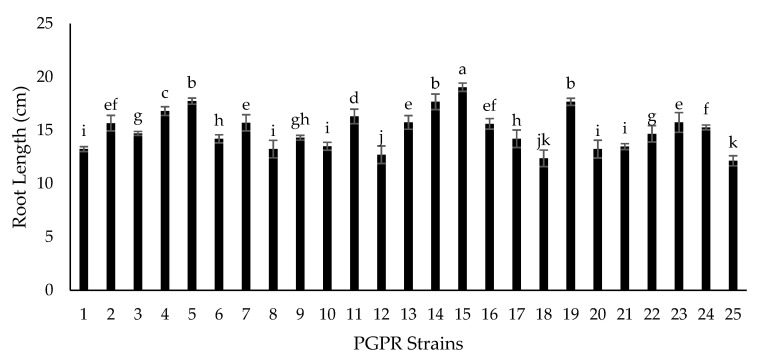
Effect of plant growth promoting rhizobacteria (PGPR) strains on chickpea root length (cm). Strains (1–24) are PGPRs (RS-01 to RS-24) while 25 is the control treatment. The bars indicate standard error (±SE) of the mean (*n* = 3). Means followed by the same letter within the bars are not significantly different at *p* = 0.05.

**Figure 9 plants-09-01629-f009:**
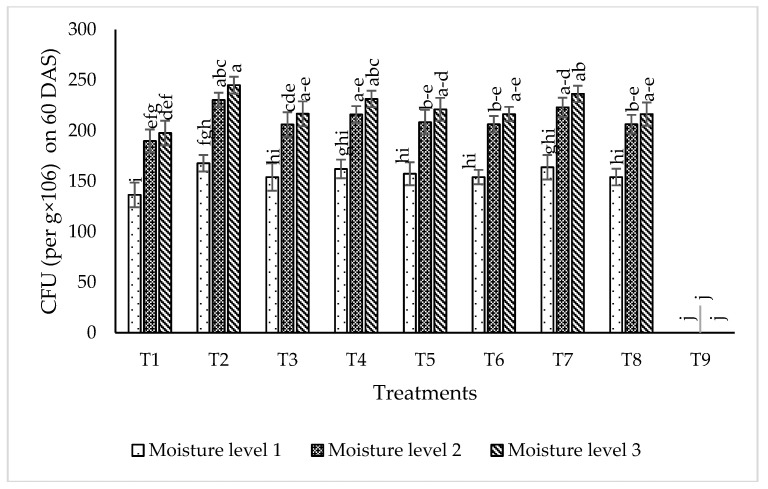
Survival of isolates Colony Forming Unit (CFU/g × 10^6^) in rhizospheric soil 60 days after sowing (DAS) at different moisture levels in glass house experiment. The bars indicate standard error (±SE) of the mean (*n* = 3). Means followed by the same letter within the bars are not significantly different at *p* = 0.05. T1 = RZ22 + RP01 + RS15, T2 = RZ11 + RP01 + RS14, T3 = RZ11 + RP08 + RS14, T4 = RZ11 + RP08 + RS15, T5 = RZ11 + RP01 + RS15,T6 = RZ22 + RP08 + RS14, T7 = RZ22 + RP08 + RS15, T8 = RZ22 + RP01 + RS14, T9 = control, where RZ11 = *Mesorhizobium ciceri*, RP08 = *Enterobacter cloacae*, RS14 = *Bacillus mojavensis*, RS15 = *Providencia vermicola*, RP01 = *Bacillus subtilis*, RZ22 = *Mesorhizobium ciceri*, Moisture level 1 = 55% of Field Capacity, Moisture level 2 = 75% of Field Capacity, Moisture level 3 = 95% of Field Capacity.

**Figure 10 plants-09-01629-f010:**
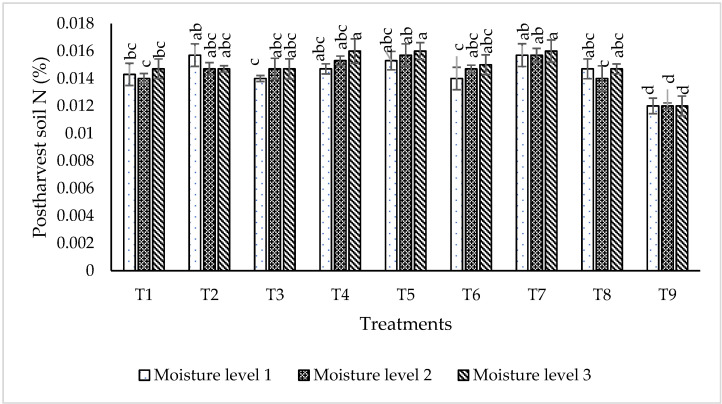
Effect of most efficient strains on post-harvest soil N (%) at different moisture levels in glass house experiment. The bars indicate standard error (± SE) of the mean (*n* = 3). Means followed by the same letter within the bars are not significantly different at *p* = 0.05. T1 = RZ22 + RP01 + RS15, T2 = RZ11 + RP01 + RS14, T3 = RZ11 + RP08 + RS14, T4 = RZ11 + RP08 + RS15, T5 = RZ11 + RP01 + RS15,T6 = RZ22 + RP08 + RS14, T7 = RZ22 + RP08 + RS15, T8 = RZ22 + RP01 + RS14, T9 = control, where RZ11 = *Mesorhizobium ciceri*, RP08 = *Enterobacter cloacae*, RS14 = *Bacillus mojavensis*, RS15 = *Providencia vermicola*, RP01 = *Bacillus subtilis*, RZ22 = *Mesorhizobium ciceri*, Moisture level 1 = 55% of Field Capacity, Moisture level 2 = 75% of Field Capacity, Moisture level 3 = 95% of Field Capacity.

**Figure 11 plants-09-01629-f011:**
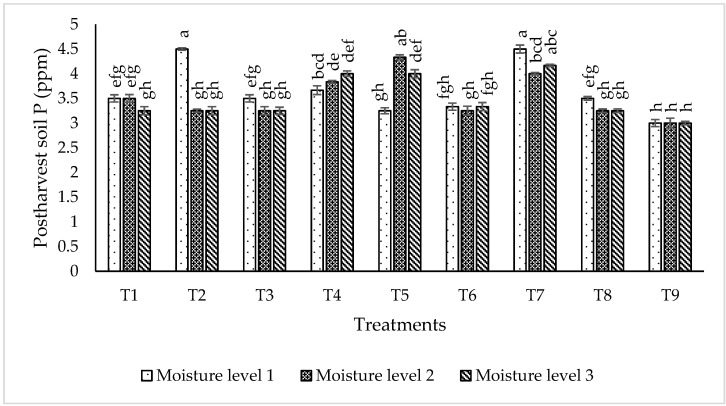
Effect of most efficient strains on post-harvest soil P (ppm) at different moisture levels in glass house experiment. The bars indicate standard error (± SE) of the mean (*n* = 3). Means followed by the same letter within the bars are not significantly different at *p* = 0.05. T1 = RZ22 + RP01 + RS15, T2 = RZ11 + RP01 + RS14, T3 = RZ11 + RP08 + RS14, T4 = RZ11 + RP08 + RS15, T5 = RZ11 + RP01 + RS15,T6 = RZ22 + RP08 + RS14, T7 = RZ22 + RP08 + RS15, T8 = RZ22 + RP01 + RS14, T9 = control, where RZ11 = *Mesorhizobium ciceri*, RP08 = *Enterobacter cloacae*, RS14 = *Bacillus mojavensis*, RS15 = *Providencia vermicola*, RP01 = *Bacillus subtilis*, RZ22 = *Mesorhizobium ciceri*, Moisture level 1 = 55% of Field Capacity, Moisture level 2 = 75% of Field Capacity, Moisture level 3 = 95% of Field Capacity.

**Table 1 plants-09-01629-t001:** Morphological characterization of most efficient isolated bacterial strains after screening.

Isolate Strain	Identified Name	Accession No.	Sample Collection Site	Gram Reaction	Bacterial Colony Shape	Form	Elevation	Color	Margin
RP-08	*Enterobacter cloacae*	MN599995	Hayderabad Thal	−ve	Round	Smooth	Raised	Yellowish	Entire
RP-01	*Bacillus subtilis*	MN599716	Mankaira	+ve	Irregular	Dry/rough	Umbonate	Yellowish	Undulate
RS-15	*Providencia vermicola*	MN601273	Rahdari	−ve	Round	Moist/Round	Convex	Light yellow	Entire
RS-14	*Bacillus mojavensis*	MN599703	Pulses research farm Kaloor kot	−ve	Round	Moist	Convex	Yellowish	Undulate
RZ-11	*Mesorhizobium ciceri*	MN601357	AZRI Bhakker	−ve	Round	Sticky	Raised	White	Entire
RZ-22	*Mesorhizobium ciceri*	MN601741	Chobara	−ve	Round	Sticky	Raised	White	Entire

**Table 2 plants-09-01629-t002:** Characterization (Biochemical parameters) of most efficient isolated bacterial strains after screening.

Isolate Strain	Isolation From	IAA Test	HCN Test	Siderophore Test	Phosphate Solubilization	Ammonia Test	Amylase Test	Protease Test	Catalase Test	ACC-Deaminase	EPS
RP-08	Rhizoplane soil	+++	−	+	+++	−	−	+	+	+	+
RP-01	Rhizoplane soil	++	−	++	+++	−	+	+	+	+	+
RS-15	Rhizoplane soil	+++	−	++	−	−	+	+	+	+	+
RS-14	Rhizoplane soil	++	−	+	−	−	+	+	+	+	+
RZ-11	Nodules	−	−	−	−	+++	−	−	−	−	−
RZ-22	Nodules	−	−	−	−	+++	−	−	−	−	−

+ Shows weak positive, ++ shows moderate positive, +++ represent strong positive, while − shows negative result for production. IAA-Indole-3-acetic acid; HCN-Hydrogen cyanide; ACC-1-aminocyclopropane-1-carboxylate; EPS-Exopolysaccharide.

**Table 3 plants-09-01629-t003:** Quantitative estimation of EPS and IAA production and ACC-deaminase and Phosphate solubilization activities.

Isolate Strain Code/Treatment	Bacterial Strains	ACC-Deaminase (μM/mg Protein/h)	Exopolysaccharide (mg/mL)	IAA Production (μg/mL)	Phosphate Solubilization (ug/L)
RP-08	*Enteroabacter cloacae*	0.84 ± 0.016	0.80 ± 0.030	86 ± 1.99	13.4 ± 0.89
RP-01	*Bacillus subtilis*	0.77 ± 0.024	0.74 ± 0.032	79 ± 1.60	14.2 ± 0.68
RS-15	*Providencia vermicola*	0.69 ± 0.012	0.68 ± 0.038	71 ± 1.80	ND
RS-14	*Bacillus mojavensis*	0.66 ± 0.022	0.61 ± 0.062	63 ± 1.40	ND
RZ-11	*Mesorhizobium ciceri*	ND	ND	ND	ND
RZ-22	*Mesorhizobium ciceri*	ND	ND	ND	ND
T1	RZ22 + RP01 + RS15,	1.3 ± 0.012	NT	121 ± 2.30	14.2 ± 0.68
T2	RZ11 + RP01 + RS14,	2.6 ± 0.011	NT	177 ± 2.60	14.2 ± 0.68
T3	RZ11 + RP08 + RS14,	1.5 ± 0.011	NT	138 ± 1.50	13.4 ± 0.89
T4	RZ11 + RP08 + RS15,	2.4 ± 0.015	NT	164 ± 2.10	13.4 ± 0.89
T5	RZ11 + RP01 + RS15	2.1 ± 0.017	NT	146 ± 2.20	14.2 ± 0.68
T6	RZ22 + RP08 + RS14	1.4 ± 0.019	NT	133 ± 1.60	13.4 ± 0.89
T7	RZ22 + RP08 + RS15	2.5 ± 0.014	NT	171 ± 2.10	13.4 ± 0.89
T8	RZ22 + RP01 + RS14	1.5 ± 0.011	NT	137 ± 1.70	14.2 ± 0.68

ND = Not Detected, NT = Not Tested.

**Table 4 plants-09-01629-t004:** Effect of inoculants on growth attributes of chickpea at different moisture regimes.

Treatment (Consortium)	Moisture Level	No. Nodules Plant^−1^	Plant Height (cm)	No. Pods Pot^−1^	Root Length (cm)	100 Grain Weight (g)	Biological Yield Pot^−1^ (g)	Economic Yield Pot^−1^ (g)	Harvest Index %	Proline (mg g^−1^ dw)
T1	1	5.00 ij	29.24 hi	17.00 ij	27.23 jkl	26.93 ij	33.20 m	14.20 kl	39.33 cd	3.7533 hij
	2	7.67 fgh	35.84 efg	19.33 fg	34.03 d–h	28.20 h	45.00 fg	16.17 ij	38.33 d	3.6167 jkl
	3	9.44 e	41.05 bcd	21.67 d	39.24 abc	29.00 d–g	49.17 cd	19.00 c–f	39.67 bcd	3.4867 lm
T2	1	12.33 d	32.37 gh	20.67 de	29.61 ijk	29.83 abc	38.93 j	17.90 fgh	44.67 ab	4.7967 a
	2	23.11 a	37.59 def	28.00 a	42.03 ab	28.37 gh	45.01 fg	20.08 bc	44.00 abc	4.2633 de
	3	22.45 a	44.07 ab	24.33 c	36.35 c–f	29.10 c–g	49.37 c	20.57 b	44.00 abc	4.1000 ef
T3	1	4.11 j	28.22 i	16.67 ijk	26.77 kl	25.67 lm	31.60 no	14.43 kl	43.67 abc	3.6833 ijk
	2	7.11 fgh	35.20 efg	18.33 gh	32.64 ghi	28.70 e–h	44.00 gh	18.85 d–g	43.00 a–d	3.5400 kl
	3	8.56 ef	40.78 bcd	20.67 de	38.40 abc	29.20 b–f	48.83 cd	20.10 bc	42.67 a–d	3.4733 lm
T4	1	6.78 gh	29.14 hi	19.67 ef	27.39 jkl	26.63 ijk	36.47 k	16.47 i	43.33 a–d	4.5233 bc
	2	19.00 b	42.80 ab	24.33 c	39.23 abc	28.57 fgh	47.53 e	20.47 b	45.33 a	4.4033 cd
	3	23.89 a	45.83 a	25.67 b	42.57 a	29.90 ab	52.07 a	22.70 a	43.33 a–d	4.2700 de
T5	1	5.00 ij	29.80 hi	17.33 hi	27.04 kl	29.33 b–e	34.37 lm	14.80 k	43.00 a–d	4.0900 ef
	2	11.89 d	41.43 bcd	23.33 c	38.03 bcd	29.87 ab	45.00 fg	19.90 bcd	43.00 a–d	3.9233 fgh
	3	11.56 d	36.13 efg	19.67 ef	33.28 e–h	29.63 a–d	48.00 de	19.73 bcd	43.00 a–d	3.7167 ijk
T6	1	3.78 j	26.79 ij	16.33 ijk	26.03 kl	24.37 n	30.73 o	13.60 l	40.00 bcd	3.9733 fg
	2	6.11 hi	34.14 fg	18.33 gh	32.95 f–i	25.97 klm	45.00 fg	19.00 c–f	41.67 a–d	3.7533 hij
	3	7.56 fgh	38.40 cde	18.33 gh	37.13 cde	27.23 i	49.80 bc	18.00 fgh	41.00 a–d	3.6267 jkl
T7	1	6.11 hi	28.40 hi	18.33 gh	28.33 jkl	28.57 fgh	35.20 l	15.30 jk	43.33 a–d	4.6067 b
	2	23.56 a	43.53 ab	29.00 a	42.47 a	29.43 a–e	52.87 a	23.17 a	41.67 a–d	4.5133 bc
	3	14.56 c	42.04 abc	24.33 c	36.87 c–f	30.13 a	50.00 bc	19.87 bcd	42.00 a–d	4.1733 e
T8	1	4.00 j	27.78 ij	16.67 ijk	26.30 kl	24.00 n	32.00 n	14.17 kl	41.33 a–d	3.8600 ghi
	2	6.67 gh	34.39 fg	18.67 fg	31.27 hij	25.36 m	46.13 f	19.30 cde	39.67 bcd	3.3333 mn
	3	8.22 efg	40.40 bcd	18.33 gh	36.24 c–f	26.30 jkl	50.83 b	18.30 efg	41.67 a–d	3.2500 n
T9	1	1.22 k	22.43 k	16.00 jk	24.70 l	23.07 o	26.49 p	11.92 m	43.00 a–d	1.4067 o
	2	1.44 k	24.07 jk	15.67 k	24.90 l	24.53 n	40.73 i	17.10 hi	42.67 a–d	1.2533 op
	3	1.44 k	24.06 jk	16.00 jk	25.13 l	25.73 lm	43.37 h	17.80 gh	42.67 a–d	1.1533 p
LSD (0.05)		1.63	4.00	1.31	4.17	0.77	1.17	1.14	5.15	0.1845

All the treatments sharing common letter are similar; otherwise, they differ significantly at *p* ≤ 0.05, T1 = RZ22 + RP01 + RS15, T2 = RZ11 + RP01 + RS14, T3 = RZ11 + RP08 + RS14, T4 = RZ11 + RP08 + RS15, T5 = RZ11 + RP01 + RS15,T6 = RZ22 + RP08 + RS14, T7 = RZ22 + RP08 + RS15, T8 = RZ22 + RP01 + RS14, T9 = control, where RZ11 = *Mesorhizobium ciceri*, RP08 = *Enterobacter cloacae*, RS14 = *Bacillus mojavensis*, RS15 = *Providencia vermicola*, RP01 = *Bacillus subtilis*, RZ22 = *Mesorhizobium ciceri*, Moisture level 1 = 55% of Field Capacity, Moisture level 2 = 75% of Field Capacity, Moisture level 3 = 95% of Field Capacity. Means followed by the same letter within a column are not significantly different at *p* = 0.05.

**Table 5 plants-09-01629-t005:** Effect of inoculants on nutrient acquisition of chickpea at different moisture regimes.

Treatment (Consortium)	Moisture Level	N in Frain (%)	P in Grain (%)	Protein in Grain (%)	N-Contents in Shoot (%)	P-Contents in Shoot (%)
T1	1	4.0600 l	0.2967 gh	25.377 hi	1.5267 f	0.2267 jkl
	2	4.0867 kl	0.3033 fgh	25.543 ghi	1.5300 ef	0.2300 jkl
	3	4.1233 hij	0.3100 ef	25.770 b–h	1.5400 def	0.2400 hij
T2	1	4.1867 b	0.3133 def	26.167 bc	1.6633 a	0.2567 gh
	2	4.1767 bcd	0.3033 fgh	26.107 bcd	1.6533 ab	0.2867 def
	3	4.1833 bc	0.3167 cde	26.150 bc	1.6300 a–e	0.3167 b
T3	1	4.0900 k	0.3067 efg	25.563 f–i	1.5300 ef	0.2300 jkl
	2	4.1200 hij	0.3067 efg	25.750 c–h	1.5567 b–f	0.2567 gh
	3	4.1667 b–e	0.3233 bcd	26.043 b–e	1.5800 a–f	0.2800 ef
T4	1	4.1067 jk	0.3033 fgh	25.667 e–h	1.6567 ab	0.2433 hij
	2	4.1367 f–i	0.3067 efg	25.857 b–g	1.6467 abc	0.2767 ef
	3	4.3133 a	0.3333 b	26.960 a	1.6367 a–d	0.3367 a
T5	1	4.1133 ijk	0.3067 efg	25.710 d–h	1.5300 ef	0.2300 jkl
	2	4.1567 c–f	0.3233 bcd	25.980 b–f	1.5500 c–f	0.2500 hi
	3	4.1700 b–e	0.3233 bcd	26.063 b–e	1.5833 a–f	0.2833 def
T6	1	4.1267 g–j	0.3033 fgh	25.790 b–h	1.5367 def	0.2367 ijk
	2	4.1533 d–g	0.3133 def	25.960 b–g	1.5700 a–f	0.2700 fg
	3	4.1767 bcd	0.3233 bcd	26.107 bcd	1.5733 a–f	0.2933 cde
T7	1	4.1333 f–j	0.3133 def	25.833 b–g	1.5833 a–f	0.2400 hij
	2	4.2867 a	0.3533 a	26.793 a	1.6533 ab	0.3000 bcd
	3	4.1900 b	0.3267 bc	26.190 b	1.6100 a–f	0.3100 bc
T8	1	4.1433 e–h	0.2967 gh	25.900 b–g	1.5333 ef	0.2333 ijk
	2	4.1700 b–e	0.3167 cde	26.063 b–e	1.5567 b–f	0.2567 gh
	3	4.1733 bcd	0.3233 bcd	26.083 b–e	1.5867 a–f	0.2867 def
T9	1	3.8900 m	0.2933 h	24.313 j	1.3100 g	0.2000 m
	2	3.8967 m	0.2933 h	24.353 j	1.3233 g	0.2133 lm
	3	3.9167 m	0.3033 fgh	25.147 i	1.3233 g	0.2200 kl
LSD (0.05)		0.77	0.0103	0.4249	0.1012	0.0183

P = Phosphorus, N = Nitrogen, T1 = RZ22 + RP01 + RS15, T2 = RZ11 + RP01 + RS14, T3 = RZ11 + RP08 + RS14, T4 = RZ11 + RP08 + RS15, T5 = RZ11 + RP01 + RS15,T6 = RZ22 + RP08 + RS14, T7 = RZ22 + RP08 + RS15, T8 = RZ22 + RP01 + RS14, T9 = control, where RZ11 = *Mesorhizobium ciceri*, RP08 = *Enterobacter cloacae*, RS14 = *Bacillus mojavensis*, RS15 = *Providencia vermicola*, RP01 = *Bacillus subtilis*, RZ22 = *Mesorhizobium ciceri*, Moisture level 1 = 55% of Field Capacity, Moisture level 2 = 75% of Field Capacity, Moisture level 3 = 95% of Field Capacity. Means followed by the same letter within a column are not significantly different at *p* = 0.05.

**Table 6 plants-09-01629-t006:** Detail set of treatments in glass house experiment on chickpea.

Treatments	Isolated Strains	Identified Bacteria
T_1_	RZ-22+RP-01+RS-15	*Mesorhizobium ciceri* RZ-22 + *Bacillus subtilis* RP-01 + *Providencia vermicola* RS-15
T_2_	RZ-11+RP-01+RS-14	*Mesorhizobium ciceri* RZ-11 + *Bacillus subtilis* RP-01 + *Bacillus mojavensis* RS-14
T_3_	RZ-11+RP-08+RS-14	*Mesorhizobium ciceri* RZ-11 + *Enterobacter cloacae* RP-08 + *Bacillus mojavensis* RS-14
T_4_	RZ-11+RP-08+RS-15	*Mesorhizobium ciceri* RZ-11 + *Enterobacter cloacae* RP-08 + *Providencia vermicola* RS-15
T_5_	RZ-11+RP-01+RS-15	*Mesorhizobium ciceri* RZ-11 + *Bacillus subtilis* RP-01 + *Providencia vermicola* RS-15
T_6_	RZ-22+RP-08+RS-14	*Mesorhizobium ciceri* RZ-22 + *Enterobacter cloacae* RP-08 + *Bacillus mojavensis* RS-14
T_7_	RZ-22+RP-08+RS-15	*Mesorhizobium ciceri* RZ-22 + *Enterobacter cloacae* RP-08 + *Providencia vermicola* RS-15
T_8_	RZ-22+RP-01+RS-14	*Mesorhizobium ciceri* RZ-22 + *Bacillus subtilis* RP-01 + *Bacillus mojavensis* RS-14
T_9_	Control	

**Table 7 plants-09-01629-t007:** Pre-sowing analysis of soil used in the glass house experiment.

Characteristics	Value	Characteristics	Value
Sand (%)	65.3	Electrical Conductivity (dSm^−1^)	0.45
Silt (%)	20	Available P (mg/kg)	3.29
Clay (%)	14.7	Available K (mg/kg)	68
Texture	Sandy loam	Organic Matter (%)	0.25
Ph	8.3	Nitrogen (%)	0.013

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
