# Peer review of "Desert Soil Microbes as a Mineral Nutrient Acquisition Tool for Chickpea (Cicer arietinum L.) Productivity at Different Moisture Regimes"

_plants, 2020, doi:10.3390/plants9121629_

Round 1

Reviewer 1 Report

The manuscript entitled “Desert Soil Microbes as Mineral Nutrient Acquisition tool for Chickpea (Cicer arietinum L.) Productivity at 4 Different Moisture Regimes” by Aulakh et al. describes the isolation and characterization of nitrogen fixing rhizobia and PGPR from desert soil and the further establishment of consortia, which were analyzed in terms of growth and development of chickpea in desert soil. The study is complete and properly formulated in terms of isolates characterization and outputs in glass house experiments; it constitutes a good incentive for further field experiments and future biofertilizer formulations for chickpea yield improvement in desert soil, which will be of main relevance for Pakistan, the 4th largest chickpea producing country. However, there are important flaws in terms of experimental work and manuscript preparation.

Three important points are missing from the manuscript:

-At line 173, it is only stated “Bacterial strain consortia were tested for their efficacy to improve productivity of chickpea …”. It is not explained how the consortia were build, for instance, where the cells of each strain collected at the stationary phase of growth? In what medium where the strains mixed? What was the volume used for each strain in the mix? Where the consortia inoculated immediately after sowing?

-Before section 2.7, three sections should be included relative to the characterization of the consortia before inoculation, which was not performed. This implies, of course, additional experiments 1) Consortia should be tested before performing the glass house experiment for compatibility between strains, 2) It would be important to  expose the presence of the bacteria after inoculation (number each strain within the consortium/seed or /g soil after a given time (x)  following inoculation , as well as the confirmation of rhizosphere colonization (nodules plant-1 after x time) to corroborate the conclusions 3) the characterization of the consortia in terms of the PGPR characteristics. For instance in line 411”.., by inoculation with T2-consortia having ACCD, EPS, IAA and..”, it cannot be said that T2 has the named abilities, as they were not tested in the consortium, but in the isolated strains. I advise the authors to follow the examples in the manuscript from Molina-Romero et al., 2017 https://doi.org/10.1371/journal.pone.0187913

Results of 1) to 3) should be, of course, mentioned in the results: Results from 1) and 2) should be exposed before section 3.7. Results from 3 should be included in tables 3 and 4 and explained in the text.

-Although the work exposed seems well done, there are many incongruences between the text and tables (see below). Also, there are misplaced references. Finally, I advise the authors to invite a native English speaker to review the paper. Writing a paper with many English mistakes is very distracting for the reviewer who needs to focus on scientific criteria for evaluation…I pointed below some corrections I believe will help the authors to advance to a posterior submission.

Title:

“Desert Soil Microbes as a Mineral Nutrient Acquisition Tool for Chickpea (Cicer arietinum L.) Productivity at Different Moisture Regimes “

In Abstract:

Line 18 “The current study was designed to illuminate…”

Line 19 “..growth and net-return of chickpeas grown in pots by using sandy loam soil of Thal Pakistan desert.”

Line 20 “A total of 125 rhizobacterial strains were isolated…”

Line 24  “Then, eight consortia of the identified..”

Line 26 “…levels (55%, 75% and 95% of field capacity) in a glass house experiment.”

Line 30”… that the consortium T2 (Mesorhizobium ciceri RZ-11 + Bacillus subtilis RP-01+ Bacillus mojavensis RS-14)

In Introduction:

Line 40 “Thus, for maintaining global food security, it is very important …”

Line 43 “..more than 9 billion by 2050 and food availability is imperative, thus the need to focus on…”

Line 47 “.. for growth-promoting representative strains to be used for …”

Line 50 ” The total area under chickpea cultivation in the world..”

Line 53 “Use of efficient bacteria in the leguminous crops for acquisition…”

Line 58 Remove the sentence: “Plant growth promoting rhizobacteria have the ability to solubilize organic phosphorus for plant growth and development.”

Line 59 “Phosphorus comes after nitrogen for nutritional requirements of plants and is one of the most important minerals required for plant growth occupying a strong position among soil macro nutrients [13].”

Line 61 Insert here part of the sentence from line 73, as it follows: “Phosphorus plays a key role in root growth and development, stimulating cell division or cell elongation.”

Line 62 “Due to phosphorus availability, the surface area, volume and root length increases, then indirectly promotes water absorption from untapped soil profile.”

Line 63. “However, in drought conditions, P-accumulation or fixation occurs due to phosphatase enzyme inactivity.”

Line 65 “..solubilizing microbes release phosphatase enzyme, which…”

Line 66 “…solubilization from fixed soil P-pools for plants P acquisition [15,16]. Thus, phosphate solubilizing microbes are a major solution for mineralization of soil fixed P in drought conditions [17].”

NOTE: Enzymes released from soil microorganisms that persists in soils are very important in mineralization. Also, chemical reactions account for P mineralization, therefore the replacement of “only by “a major”.

Line 68 “Moreover, about 80 % of applied phosphorus in fertilizers gets fixed in soils.”

Line 69 ”… metal oxides, and soil pH are also main factors…”

Line 71 “Thus, PGPR (Plant Growth Promoting Rhizobacteria) having P-solubilizing activity, mineralizing insoluble phosphorus into soluble form by producing organic acids, contribute to a massive increase in the crop yield [20].”

Line72 (to 75) Remove the sentences: “Moreover, phosphorus plays a key role in root growth and development [16]. Drought stress is a major obstacle in high yield of chickpea crop in Thal Desert, Pakistan. Extensive root growth of plants may be helpful for water absorption from untapped soil profiles.” Replace by: “Considering the key role of phosphorus in root growth and development [16], an increase in its availability and consequent potential extensive root growth of plants will be helpful for water absorption from untapped soil profiles, an important goal for high yield of chickpea crop in Thal Desert, Pakistan, where drought stress is a major obstacle.

Line 76   Remove  the sentence (see below line 89): “ Plant Growth Promoting Rhizobacteria (PGPR) have significant positive response to alleviate 77 drought stress in dry land crops by producing ACC-deaminase, exopolysaccharides (EPS) and 78 phytohormones in acquisition of drought tolerance in plants [21].

Line 81“… depends upon annual rainfall as well as on some patches of the area under tube well irrigation system.”

Line 88 “..deserts, the present study was planned to estimate..”

Line 89 Introduce the following sentence before “The hypothesis…”: “ It has been reported that PGPR have significant positive effects in alleviating drought stress in dry land crops by producing ACC-deaminase, exopolysaccharides (EPS) and phytohormones, contributing in the acquisition of drought tolerance in plants [21].The hypothesis of our study is that isolated bacteria may be helpful to increase the drought tolerance in plants by secreting different compounds like ABA, exopolysaccharides, producing ACC-deaminase and enzymatic activities. “

Line 93 NOTE: At the end of the introduction, the purpose of building consortia should be mentioned. For instance: “Building consortia between PGPR, including those having P-solubilizing activity and rhizobium species might increase significantly chickpea crop production.”

In Material and methods:

Line 104 “and colonies were grown on plates after incubation at 28 ± 2 104 0C for 7 days. PGPR…”

Line 107 (see below line 197): Insert in a different paragraph,  after “..at 28 ± 2 107 0C for 24 to 48 hours “, the following:  “ A total of 125 rhizobacterial strains were isolated from root nodules (RZ), rhizospheric (RS) and  rhizoplane (RP) soil of chickpea plants. Among them, 72 were repeatedly streaked on respective mediums in petri plates to obtain pure colonies for a screening process in growth chamber.”

Line 108 “2.3. Morpho-physiological characterization of isolated strains”

Line 116 “…production by following the (give the name of the method) method [27]”

NOTE: Check the references, ammonia production is not [27], Also, in line 127, check references [28] (should be [27] and [29].

Line 118 “Phosphate solubilization efficiency and index of isolates was determined by the method of Macfaddin [30].”

Line 123 “Bacterial strains were tested for catalase production using the procedure given by [31].

Line 126 “…of catalase enzyme. For the amylase test, iodine solution …”

Line 129 “…adopting the method described in [32]. NOTE: Here again, check the references.

Line 130 “ Then, a loop of the bacterial strain  suspended in (insert the solution where the bacteria were suspended) was shifted onto sterile DF (Dworkin and Foster) salt media containing ACC as a single source of nitrogen. Afterwards, the plates with the bacterial strains were incubated for 3 days at 28 ºC and checked for colony growth showing ACC deaminase activity [34].

Line 134 “…as nmol α-ketobutyrate mg protein-1 hr -1 using a spectrophotometer at 540 nm wavelength [35].

Line 135 “…were characterized by streaking them on ATCC medium no.14; plates were incubated for 3 days at 111 0C (????? Check the temperature, probably it was 28ºC). The bacterial colonies showing slimes around them were characterized as EPS producing strains [35]. NOTE: Here again, check the references.

Line 140 “…Agriculture Research Centre, Islamabad to evaluate the effect of isolates, 24 Rhizobium (RZ-01 to RZ-24) and 48 Plant Growth Promoting Rhizobacteria on growth attributes of chickpea seedling. Insert here the sentence:  The chosen PGPRs were isolated from rhizoplane soil (RP-01 to RP-24) and from rhizospheric soil (RS-01 to RS-24) so to screen out the 2 best strains from each category.” NOTE: Following this sentence and considering the Figure 5 and Figure 6 (see line 270 below), a sentence should be included to describe the PSB character of RP-01 to RP-24 strains.

Line 144 “…isolates were sown. Uninoculated seeds were sown in the jar..”

Line 145 “..control treatment. These jars were kept in the growth chamber in a complete randomized design.

Line 147 “Among the isolates, two best Rhizobium and four PGPRs were chosen on the basis …”

Line 148 “…on growth attributes of chickpea. The chosen isolates were molecularly characterized and identified..”

Line 149 “…consortium effects on the productivity of chickpea at different moisture regimes.”

Line 154 “Purified colonies of the most efficient bacterial strains were plucked and mixed with 20μL Tris-EDTA buffer in Polymerase Chain Reaction (PCR) strips.

Line 156 NOTE: Not all laboratories have a PCR apparatus allowing centrifugation. Thus, the name of the apparatus should be given. The description of the PCR preparation and procedure is not clear and must be improved:  “This mixture was centrifuged in a PCR apparatus (give name) for 10 minutes at 95Co (check temperature of centrifugation,  10 min and 95ºC are usually the conditions for DNA denaturation, following centrifugation. ) and the supernatant was collected as DNA template. DNA amplification was carried out by using 2μL of forward and reverse universal primers 9F (5; 157 - GAGTTGATCCTGGCTCAG-3;) and 1510R (5;-GGCTACCTTGTTACGA-3; 158 ), respectively, 25μL TAKARA Pre-mix Ex-Taq, 20μL PCR water and 1μL of DNA template. The amplified PCR products were sent to Macrogen, Korea, for sequencing and strains were identified using the EzBioCloud server.”

Line 165 “…and yield attributes of chickpea in comparison to the untreated control. For the experiment…”

Line 169 “…of formula in [37,38] and the moisture was maintained with the help of a Time Domain Reflectometer (TDR) used during the experiment. ..”~

Line 171 NOTE: After the sentence “.. (both inoculated and uninoculated) were sown in respective pots.”, it should be mentioned if the experiment follows with all the germinated seeds, or if only one plant per pot is maintained.

Line 174: NOTE: “…for their efficacy to improve productivity of chickpea with the following set of treatments:”

Line 175, footnote of Table, remove “Where RZ-11 (Mesorhizobium ciceri) = collected from research farm of Arid Zone Research Institute, Bhakker and RZ-22 (Mesorhizobium ciceri) = collected from Chobara, district Layyah.”, because this is exposed further in Table 2.

Line 181 “..in table 1. The post-harvest soil properties (Total Nitrogen and Extractable phosphorus) were determined by adopting the methods described in [42] and [43], respectively.

Line 184, title of table “Table 1. Pre-sowing analysis of soil used in the in vitro experiment.”

Line 186 “Proline contents in 130 days old leaves were determined following the method described in [45].

Line 189 “..upper pink layer was selected for quantification using a spectrophotometer at 520 nm wavelength.”

Line 190 “Seed protein contents were analyzed following the method described in [42].”

In Results:

Line 197 NOTE. Section 3.1. should be removed and the information passed to “Materials and methods” section. After correcting line 199 below as suggested, move the corrected sentence to section 2.2., as explained above for line 107.

Line 199 “Among them, 72 were repeatedly streaked on respective mediums in petri plates to obtain pure colonies for a screening process in growth chamber.” i.e., remove “different bacterial strains” and “finalized for the screening process”.

Line 203 “Initially, 72 strains were tested for screening and among them, the 6 most promising strains were observed by Phase Contrast Microscope (Phase contrast 2, Nikon, Japan) for the colony morphology. MIRA3, Tescan SEM (scanning electron microscope) was used to characterize (insert here which characteristics) (Table 2).” NOTE: the last 5 descriptions in table 2 refer to colony morphology, so it is relevant to mention which of the characters were analyzed with MIRA.

Line 206 Remove the paragraph “The only RP-01colony showed umbonate elevation with yellow color and marked as gram +ve 207 instead of all other strains that were morphologically gram –ve. The strain RP-08 (Enterobacter cloacae) …. very sticky with raised elevation and off white in color”, since the characteristics are shown in Table 2. NOTE: There are incongruencies between the text and the table, for instance, “The only RP-01colony showed umbonate elevation with yellow color” is noted “white” in the table, please check data in the table.

Line 211, Table 2 For RS-14: Opaque is not an elevation character, please correct. Also, for RZ-11 and RZ-22, remove the word “off”.

Line 213 “Identified bacterial strains were biochemically characterized. Two strains (RZ-11 and RZ214 22) showed positive results for ammonia production, whereas others remained…”

Line 215 “..of yellowish colored zones around the colony indicated siderophore production.”

Line 216 NOTE: Again, there is an incongruence with the table, where several isolates tested positive for siderophore production, mostly RP-01 and RZ-15. Again here, check your data.

Line 217 “Regarding the HCN test, all strains tested negative. The appearance…”

Line 220 “…and weak positive, respectively, for IAA production. However, for solubilization of tri-calcium phosphate, 2 strains (RP-01…”

Line 222 “…solubilization index % (3.00 and 2.90), whereas RS-14 and RS-15 did not show P223 solubilization activity.”

Line 223 NOTE: Again, check your data,  according to the table, only RP-08 and RS-14 had both catalase and protease activity, so the sentence should be : “Moreover, 4 PGPR strains (RS-14, RS-15, RP-01, RP-08) revealed strong potential for the catalase and/or protease enzymes production and 2 strains RS-14 and RS-15 showed positive response for amylase production (Table 3). “

Line 226, Table 3 title “Characterization (Biochemical parameters) of most…”

Line 227: Remove from the table footnote “Biochemical parameters”

Line 231 “The identified nucleotides sequences were arranged in the phylogenetic tree (Fig. 1) using MEGA 7 [47]. Isolate RZ-11 matches to Mesorhizobium ciceri strains RZ-22, SS1 (5) and CM-25”. Delete “by 64%, 94% and 100% respectively.”

Line 234 “Similarly, strain RS-14 has matching criteria to bacteria Bacillus mojavensis strain RS-1, PMCC-9 and 235 LMB3G43 in the phylogenetic tree and RS-15 matches 100% to Providencia vermicola and is clustered to 3 strains Mum1, 236 Ag1 and OF6. “

Line 236:  “PGPR isolates having P-solubilizing ability coded RP-01 and RP-08 match to Bacillus subtilis strain XGL205 and Enterobacter cloacae strain MSK, respectively.”

Line 239, Figure 1 legend. “Phylogenetic tree showing the screened isolates given in bold letters and closest homologues.” Add in the figure legend what represent the numbers in the nodes. NOTE: If the numbers in the nodes are bootstrap support values, they translate the support of the monophyletic clades, therefore the proposed changes in line 231 and next lines.

Line 243 “…maximum quantity of ACCD, EPS and IAA, which production averaged 0.84 μmol mg -1 protein hr-1), 0.80 mg mL-1 and 86 μmol mL-1, respectively.” Check the units, μM/ml is not possible…Check also the symbols of SI nomenclature, e.g. L is for liter.

Line 244 “Isolate RP-01 showed the highest quantity (14.2 ug L-1) of P-solubilization ..”

Line 246, Table 4.  Check the units in the table. Explain ND in a table footnote.

Line 251 “Seventy two Isolated strains were evaluated in a trial conducted in growth chamber to screen out most efficient rhizobium and PGPR strains for different attributes of seedling growth (Fig. 3).

Line 254 “..nodules per plant, respectively (Fig. 4). Among all tested isolates, two PSB strains (RP-01 and RP- 08) significantly enhanced shoot length by 19.4 cm and 19.5 cm, respectively (Fig. 5). A similar trend was recorded ..”

Line 257 “Two PGPR isolates (RS-14 and RS-15) were found most efficient for growth and development of the plant shoot and root in screening trials. Delete “Due to strong enzymatic activities”, as this should be part of the discussion section.

Line 260 “On the other hand, the highest root length (19.033 cm) was recorded for the treatment where chickpea seeds were inoculated with RS-15 followed by RS-14 (17.667 cm).”

Line 264, Figure 3 legend. “Effect of strains on shoot length of chickpea in growth chamber, where..”

Line 267, Figure 4 legend. “Effect of Rhizobium strains on nodulation plant-1. The bars indicate standard error (±SE) of the mean (n = 3). “NOTE: See also line 372 below.

Line 270, Figure 5 legend. “Effect of PSB strains on chickpea shoot length (cm) The bars indicate standard error (±SE) of the mean (n = 3). NOTE: See also line 372 below. Also, In Figure 5 (and Figure 6), PSB strains are showed in the x-axis; this suggests that RP-01 to 24 strains are all PSB and the remained are classed as other PGPR. This is not explained in the Materials and methods section and this should be explained after the sentences suggested for line 140 above.

Line 273, Figure 6 legend. “Effect of PSB strains on chickpea root length (cm). The bars indicate standard error (±SE) of the mean (n = 3). All means are significantly …” NOTE: See also line 372 below.

Line 276, Figure 7. “Effect of PGPR strains on chickpea shoot length (cm). The bars indicate standard error (±SE) of the mean (n = 3). All means are significantly…” NOTE: See also line 372 below.

Line 279, Figure 8. “Effect of PGPR strains on chickpea root length (cm). The bars indicate standard error (±SE) of the mean (n = 3). “NOTE: See also line 372 below.

Line 284 “height (32.37 cm), number of pods plant-1 (20.67)” NOTE: pods plant-1 in the text, however per pot in the table. Thus, the correction proposed above line 171.

Line 287 “..level 1 (55% of soil field capacity) (Table 5).”

Line 293, Table 5. “Effect of inoculants on growth attributes of chickpea at different moisture regimes” NOTE: Check congruence with text. Also correct “No. pods per pot t-1 “to “No. pods pot-1

Line 298, footnote Table 5. “…RZ22= Mesorhizobium ciceri, Moisture level 1= 55% of Field Capacity, Moisture level 2= 75% of Field Capacity, Moisture level 3= 95% of Field Capacity.”

Line 303 “…mgg-1 DW) at same moisture level, while minimum proline contents (1.1533 mg g-1 DW)..NOTE: Again, check your data, as text description do not match the table..

Line 304 “determined from untreated plants at moisture level 3(95% FC). Maximum grain N and protein contents…”

Line 306 “respectively (Table 6). T2 responded most effectively at moisture level 1 (55% FC) both for grain N and protein (4.19% and 26.17%) contents. Lower N and protein contents were observed in grains of untreated plants at all given moisture levels. NOTE: Again, check your data, as text description do not match the table.

Line 310 “…moisture level 2 by showing 0.35% P contents in grain followed by T4 at moisture level 3. T2 was the best performing consortium among all treatments, including the control, at moisture level 1, which resulted in 0.31 % P in grains of chickpea.

Line 312 “..0.31 % P in grains of chickpea. The data regarding nitrogen acquisition in chickpea shoot shows that maximum nitrogen contents (1.66% - 1.65%) were recorded for treatments T2 and T4 at moisture level 1 (55% of field capacity). Similarly, T7 appeared as the most efficient consortium with 1.65% N content in chickpea shoot at moisture level 2. Contrarily, T9 (control) showed the lowest N contents (1.31% -1.32%) in shoot of chickpea at all moisture levels.

Line 317 “..the P contents in chickpea shoot (Table 6) showed the highest P (0.34%)  for T4 at moisture  level 3, followed by T2, which showed 0.32% P at the same moisture level.”

Line 319. “..at moisture level 2. T9 (control) was recorded as the lowest performing treatment, which showed 0.20 - 0.22% P contents in chickpea shoots at all given moisture regimes.

Line 321, Table 6.Effect of inoculants on nutrient acquisition of chickpea at different moisture regimes.”

Line 323 “The data regarding the nitrogen percentage in post-harvest soil (Fig.9) show that the highest  N (0.016%) was recorded for the treatments T4, T5 and T7 at moisture level 3, followed by T2 with  0.057% (NOTE: It cannot be this value, it has to be lower than 0.016%, check your data) N in post-harvest soil at moisture level 1. “Similarly, T7 gave 0.057% N at moisture level 1 and 2. T9 (control) was recorded as the lowest performing treatment, which showed 0.012% N in post-harvest soil, at all given moisture levels.

Line 326 “..level 1 and 2. T9 (control) was recorded as the lowest performing treatment, which showed  0.012% N in post-harvest soil at all given moisture levels.

Line 328 “…was recorded for both treatments T2 and T7 at moisture level 1 (55% of field capacity).

Line 329 “…(55% of field capacity). T7 was recorded as the most efficient consortium by showing 4.166 ppm post-harvest soil P at moisture level 3. T9 (control) showed the lowest P contents (3.00 ppm NOTE: here again, the number was incorrect, please check your data, figures and text have to match…) in post-harvest soil at all moisture levels.

Line 333, Figure 9. NOTE (the same note applies to Figure 10): Letters should be included over the bars to represent the significantly different means.

Line 335 “The bars indicate standard error (±SE) of the mean (n = 3

Line 337 “…T9=control, where RZ11= Mesorhizobium ciceri, …”

Line 339 “…Moisture level 1= 55% of Field Capacity, Moisture level 2= 75% of Field Capacity, Moisture levels 3= 95% of Field Capacity”

In Discussion:

Line 355 “..by [48], who showed that some soil microbes produce ammonia and enhanced crop growth attributes.

Line 355 “Two PGPR isolates, RS-14 (Bacillus mojavensis) and RS-15 (Providencia vermicola), were evaluated as IAA producing bacterial strains.”

Line 357 “Several microbes produce active auxin as IAA, which is a plant growth promoter [49,50]. “

Line 358 “…”Two PGPR isolates RP-01 (Bacillus subtilis) and RP-08 (Enterobacter cloacae) were evaluated as most …”

Line 359 “..phosphate solubilizers. The results are also in agreement with the findings of [51], who observed the maximum phosphate solubilizing ability for Enterobacter sp.”

Line 361 “..zones surrounding the microbial colonies could be due to the synthesis of organic acids with low molecular weight, or due to polysaccharides and phosphatase production by phosphate solubilizing microbes 363 [52].”

Line 368 (to 371) “We selected 24 bacterial strains each ….” Delete, as it was passed to section 2.5. (see line 140).

Line 372 “..Nodulation plant-1 372 was increased 100% by rhizobium strains (RZ-11, RZ-22) as compared …” NOTE: If 25 represents the control in Figure 4, nodulation has increased ca tenfold as compared to uninoculated plants. Also, Figures 4 to Figure 8 should explain in the legend that 25 corresponds to the non-inoculated plants.

Line 374 “..reported that Mesorhizobium ciceri inoculation increased significantly the nodulation of chickpea plants..”

Line378 “..length by 54.4 % and 54.9 % over the un-inoculated control.”

Line 380 “The root and shoot length may be increased due to increase availability of phosphorus thanks to PSBs activity, since phosphorus has an important role in root development and cell division. All plant growth promoting isolates showed significant improvement in seedling shoot length, between 6.39% and 45.82%, as compared to the control; two PGPR isolates, RS-14 and RS-15, having phytohormonal activity, increased chickpea shoot length 42.73% and 45.82% ,respectively, in comparison to the untreated control.

Line 385 “Moreover, PGPR strains (RS-01 to RS-24) showed a significantly positive response in chickpea seedlings root length (from 0.55% to 56.87%) as compared to the control.”

Line 388 “..and RS-14, respectively, over the control without inoculation.”

Line 390 “Our experimental results are also supported by findings of (Insert here name of the authors )[60], who showed that plant-root system increases up to 40% in PGPRs treated plants as compared to untreated controls.”

Line 393 “Six most efficient strains were used to make 8 different prolific consortia that were evaluated..”

Line 395 “T2 consortium (Mesorhizobium ciceri RZ-11 + Bacillus subtilis RP-01+ Bacillus mojavensis RS-14) was found most effective for chickpea growth and yield attributes at moisture level 1, which was maintained on 55 % of field capacity.”

Line 398 “..19.8% in root length, 29% in 100 grain weight, 46.9% in  biological yield pot-1, 50% in economic yield pot-1  compared to the untreated control at moisture level 1 (55 % of FC).”

Line 402 “..production by PGPRs from the consortia in the applied treatment.”

Line 405 “..with the un-inoculated treatment. Quantity of nodules, economic yield and biological yield were also increased by 73.53%, 31.76% and 24.37% respectively, over the control.”

Line 408 “..at moisture level 1 and 3 (55 % & 95 % FC), respectively, over the control. T2 resulted in postharvest soil N and P increases of 30% and 50%, respectively, in comparison to the un-inoculated treatment at 55 % of field capacity.”

Line 410 “Also in the present study, the calculated proline content of chickpea leaves showed an increasing trend at moisture level 1 (55 % of FC) by inoculation with T2-consortium having ACCD activity, and EPS and IAA production.”

Line 413 “..[62]. Statistically higher grain yield (32.76%, Please, check this value, according to the table is 35.50) was obtained with T7 treatment (Mesorhizobium ciceri RZ-22 + Enterobacter Cloacae RP-08 + Providencia vermicola RS-15) at moisture level 2 (75% of FC) and T4 (Mesorhizobium ciceri RZ-11 + Enterobacter Cloacae RP-08 + Providencia vermicola RS-15) (31%, Please, check this value, according to the table 27.53%) at moisture level 3 (95% of FC), as compared to untreated chickpea plants. NOTE: several times in the manuscript Enterobacter is misspelled!

Line 417 “Hence, microbial combinations in T4 and T7 could be used to make effective biofertilizers for chickpea growing areas under rainfed conditions to help plants to cope with

drought spells.”

Line 422 “…of drought through experimentation on desert soil is quite a novel idea being adopted in the present study.” NOTE: it is a valid idea, but not so novel…

Line 425 “..the vital role of isolated strains to be utilized as bio-fertilizers under drought spell in Thal desert, a main chickpea producing area in Pakistan, which is the 4th  largest chickpea producing country.

Line 426 “..producing country. In this study, we found that the consortium T2 (Mesorhizobium ciceri RZ-11 + Bacillus subtilis RP-01+Bacillus mojavensis 428 RS-14) can perform best in drought conditions (55% field capacity)..”

In references:

NOTE: Check your references, those must be in agreement with the text, for instance see line 116 above.

Author Response

The manuscript entitled “Desert Soil Microbes as Mineral Nutrient Acquisition tool for Chickpea (Cicer arietinum L.) Productivity at 4 Different Moisture Regimes” by Aulakh et al. describes the isolation and characterization of nitrogen fixing rhizobia and PGPR from desert soil and the further establishment of consortia, which were analyzed in terms of growth and development of chickpea in desert soil. The study is complete and properly formulated in terms of isolates characterization and outputs in glass house experiments; it constitutes a good incentive for further field experiments and future biofertilizer formulations for chickpea yield improvement in desert soil, which will be of main relevance for Pakistan, the 4th largest chickpea producing country. However, there are important flaws in terms of experimental work and manuscript preparation.

Comment 1 Three important points are missing from the manuscript:

-At line 173, it is only stated “Bacterial strain consortia were tested for their efficacy to improve productivity of chickpea …”. It is not explained how the consortia were build, for instance, where the cells of each strain collected at the stationary phase of growth? In what medium where the strains mixed? What was the volume used for each strain in the mix? Where the consortia inoculated immediately after sowing?

Response: Information regarding the question has been incorporated in the manuscript at line 204-212 and page no 5 and 216-221 (page 6) in red colour.

Comment 2 -Before section 2.7, three sections should be included relative to the characterization of the consortia before inoculation, which was not performed. This implies, of course, additional experiments

1) Consortia should be tested before performing the glass house experiment for compatibility between strains,

Response: As per recommendation, the point-1 for compatibility has been incorporated at line no 181-188 on page 4 along with line no 355 at page 13

 2) It would be important to  expose the presence of the bacteria after inoculation (number each strain within the consortium/seed or /g soil after a given time (x)  following inoculation , as well as the confirmation of rhizosphere colonization (nodules plant-1 after x time) to corroborate the conclusions

Response: The presence of bacteria in rhizospheric soil has been given at line no 204-212 page 5 along with Fig-09 at line no 397-411 page no 16

3) the characterization of the consortia in terms of the PGPR characteristics. For instance in line 411”.., by inoculation with T2-consortia having ACCD, EPS, IAA and..”, it cannot be said that T2 has the named abilities, as they were not tested in the consortium, but in the isolated strains. I advise the authors to follow the examples in the manuscript from Molina-Romero et al., 2017 https://doi.org/10.1371/journal.pone.0187913 Results of 1) to 3) should be, of course, mentioned in the results: Results from 1) and 2) should be exposed before section 3.7. Results from 3 should be included in tables 3 and 4 and explained in the text.

Response: The required data has been incorporated and given at line no 310-313 page 10 (table-4).

 Comment 3: -Although the work exposed seems well done, there are many incongruences between the text and tables (see below). Also, there are misplaced references. Finally, I advise the authors to invite a native English speaker to review the paper. Writing a paper with many English mistakes is very distracting for the reviewer who needs to focus on scientific criteria for evaluation…I pointed below some corrections I believe will help the authors to advance to a posterior submission.

 Response: As per recommendations, the whole manuscript is revised and improved for linguistic and technical mistakes carefully.

Comment 4: Title:“Desert Soil Microbes as a Mineral Nutrient Acquisition Tool for Chickpea (Cicer arietinum L.) Productivity at Different Moisture Regimes “

Response: Needful changes have been incorporated according to suggestions of worthy reviewer at line no 2.

n Abstract:

Comment 5: ILine 18 “The current study was designed to illuminate…

Response: Needful changes have been incorporated according to suggestions of worthy reviewer at line no 17.”

Comment 6: Line 19 “..growth and net-return of chickpeas grown in pots by using sandy loam soil of Thal Pakistan desert.”

Response: Needful changes have been incorporated according to suggestions of worthy reviewer at line no 19-20.

Comment 7: Line 20 “A total of 125 rhizobacterial strains were isolated…”

Response: Needful changes have been incorporated according to suggestions of worthy reviewer at line no 20.

Comment 8: Line 24  “Then, eight consortia of the identified..”

Response: Needful changes have been incorporated according to suggestions of worthy reviewer at line no 24.

Comment 9: Line 26 “…levels (55%, 75% and 95% of field capacity) in glass house experiment.”

Response: Needful changes have been incorporated according to suggestions of worthy reviewer at line no 26.

Comment 10: Line 30”… that the consortium T2 (Mesorhizobium ciceri RZ-11 + Bacillus subtilis RP-01+ Bacillus mojavensis RS-14)

Response: Needful changes have been incorporated according to suggestions of worthy reviewer at line no 30.

In Introduction:

Comment 11: Line 40 “Thus, for maintaining global food security, it is very important …”

Response: Needful changes have been incorporated according to suggestions of worthy reviewer at line no 40.

Comment 12: Line 43 “..more than 9 billion by 2050 and food availability is imperative, thus the need to focus on…”

Response: Needful changes have been incorporated according to suggestions of worthy reviewer at line no 43.

Comment 13: Line 47 “.. for growth-promoting representative strains to be used for …”

Response: Needful changes have been incorporated according to suggestions of worthy reviewer at line no 48.

Comment 14: Line 50 ” The total area under chickpea cultivation in the world..”

Response: Needful changes have been incorporated according to suggestions of worthy reviewer at line no 50-51.

Comment 15: Line 53 “Use of efficient bacteria in the leguminous crops for acquisition…”

Response: Needful changes have been incorporated according to suggestions of worthy reviewer at line no 54.

Comment 16: Line 58 Remove the sentence: “Plant growth promoting rhizobacteria have the ability to solubilize organic phosphorus for plant growth and development.”

Response: Needful changes have been incorporated according to suggestions of worthy reviewer at line no 59-60.

Comment 17: Line 59 “Phosphorus comes after nitrogen for nutritional requirements of plants and is one of the most important minerals required for plant growth occupying a strong position among soil macro nutrients [13].”

Response: Needful changes have been incorporated according to suggestions of worthy reviewer at line no 61.

Comment 18: Line 61 Insert here part of the sentence from line 73, as it follows: “Phosphorus plays a key role in root growth and development, stimulating cell division or cell elongation.”

Response: Needful changes have been incorporated according to suggestions of worthy reviewer at line no 62-63.

Comment 19: Line 62 “Due to phosphorus availability, the surface area, volume and root length increases, then indirectly promotes water absorption from untapped soil profile.”

Response: Needful changes have been incorporated according to suggestions of worthy reviewer at line no 64-65.

Comment 20: Line 63. “However, in drought conditions, P-accumulation or fixation occurs due to phosphatase enzyme inactivity.”

Response: Needful changes have been incorporated according to suggestions of worthy reviewer at line no 65.

Comment 21: Line 65 “..solubilizing microbes release phosphatase enzymewhich…”

Response: Needful changes have been incorporated according to suggestions of worthy reviewer at line no 66-68.

Comment 22: Line 66 “…solubilization from fixed soil P-pools for plants P acquisition [15,16]. Thus, phosphate solubilizing microbes are a major solution for mineralization of soil fixed P in drought conditions [17].”

NOTE: Enzymes released from soil microorganisms that persists in soils are very important in mineralization. Also, chemical reactions account for P mineralization, therefore the replacement of “only by “a major”.

Response: Needful changes have been incorporated according to suggestions of worthy reviewer at line no 69.

Comment 23: Line 68 “Moreover, about 80 % of applied phosphorus in fertilizers gets fixed in soils.”

Response: Needful changes have been incorporated according to suggestions of worthy reviewer at line no 70.

Comment 24: Line 69 ”… metal oxides, and soil pH are also main factors…”

Response: Needful changes have been incorporated according to suggestions of worthy reviewer at line no 72.

Comment 25: Line 71 “Thus, PGPR (Plant Growth Promoting Rhizobacteria) having P-solubilizing activity, mineralizing insoluble phosphorus into soluble form by producing organic acids, contribute to a massive increase in the crop yield [20].”

Response: Needful changes have been incorporated according to suggestions of worthy reviewer at line no 74-75.

Comment 26: Line72 (to 75) Remove the sentences: “Moreover, phosphorus plays a key role in root growth and development [16]. Drought stress is a major obstacle in high yield of chickpea crop in Thal Desert, Pakistan. Extensive root growth of plants may be helpful for water absorption from untapped soil profiles.” Replace by: “Considering the key role of phosphorus in root growth and development [16], an increase in its availability and consequent potential extensive root growth of plants will be helpful for water absorption from untapped soil profiles, an important goal for high yield of chickpea crop in Thal Desert, Pakistan, where drought stress is a major obstacle.

Response: Needful changes have been incorporated according to suggestions of worthy reviewer at line no 78-81.

Comment 27: Line 76   Remove  the sentence (see below line 89): “ Plant Growth Promoting Rhizobacteria (PGPR) have significant positive response to alleviate 77 drought stress in dry land crops by producing ACC-deaminase, exopolysaccharides (EPS) and 78 phytohormones in acquisition of drought tolerance in plants [21].

Response: Needful changes have been incorporated according to suggestions of worthy reviewer at line no 82-85.

Comment 28: Line 81“… depends upon annual rainfall as well as on some patches of the area under tube well irrigation system.”

Response: Needful changes have been incorporated according to suggestions of worthy reviewer at line no 88.

Comment 29: Line 88 “..deserts, the present study was planned to estimate..”

Response: Needful changes have been incorporated according to suggestions of worthy reviewer at line no 94.

Comment 30: Line 89 Introduce the following sentence before “The hypothesis…”: “ It has been reported that PGPR have significant positive effects in alleviating drought stress in dry land crops by producing ACC-deaminase, exopolysaccharides (EPS) and phytohormones, contributing in the acquisition of drought tolerance in plants [21].The hypothesis of our study is that isolated bacteria may be helpful to increase the drought tolerance in plants by secreting different compounds like ABA, exopolysaccharides, producing ACC-deaminase and enzymatic activities. “

Response: Needful changes have been incorporated according to suggestions of worthy reviewer at line no 96.

Comment 31: Line 93 NOTE: At the end of the introduction, the purpose of building consortia should be mentioned. For instance: “Building consortia between PGPR, including those having P-solubilizing activity and rhizobium species might increase significantly chickpea crop production.”

 Response: Needful changes have been incorporated according to suggestions of worthy reviewer at line no 100-101.

In Material and methods:

Comment 32: Line 104 “and colonies were grown on plates after incubation at 28 ± 2 104 0C for 7 days. PGPR…”

Response: Needful changes have been incorporated according to suggestions of worthy reviewer at line no 113-114.

Comment 33: Line 107 (see below line 197): Insert in a different paragraph,  after “..at 28 ± 2 107 0C for 24 to 48 hours “, the following:  “ A total of 125 rhizobacterial strains were isolated from root nodules (RZ), rhizospheric (RS) and  rhizoplane (RP) soil of chickpea plants. Among them, 72 were repeatedly streaked on respective mediums in petri plates to obtain pure colonies for a screening process in growth chamber.”

Response: Needful changes have been incorporated according to suggestions of worthy reviewer at line no 116-115.

Comment 34: Line 108 “2.3. Morpho-physiological characterization of isolated strains”

Response: Needful changes have been incorporated according to suggestions of worthy reviewer at line no 121.

Comment 35: Line 116 “…production by following the (give the name of the method) method [27]”

NOTE: Check the references, ammonia production is not [27], Also, in line 127, check references [28] (should be [27] and [29].

Response: Needful changes have been incorporated according to suggestions of worthy reviewer at line no 129.

Comment 36: Line 118 “Phosphate solubilization efficiency and index of isolates was determined by the method of Macfaddin [30].”

Response: Needful changes have been incorporated according to suggestions of worthy- reviewer at line no 131-132.

Comment 37: Line 123 “Bacterial strains were tested for catalase production using the procedure given by [31].

Response: Needful changes have been incorporated according to suggestions of worthy reviewer at line no 136.

Comment 38: Line 126 “…of catalase enzyme. For the amylase test, iodine solution …”

Response: Needful changes have been incorporated according to suggestions of worthy reviewer at line no 139.

Comment 39: Line 129 “…adopting the method described in [32]. NOTE: Here again, check the references.

Response: Needful changes have been incorporated according to suggestions of worthy reviewer at line no 142.

Comment 40: Line 130 “ Thena loop of the bacterial strain  suspended in (insert the solution where the bacteria were suspended) was shifted onto sterile DF (Dworkin and Foster) salt media containing ACC as a single source of nitrogen. Afterwards, the plates with the bacterial strains were incubated for 3 days at 28 ºC and checked for colony growth showing ACC deaminase activity [34].

Response: Needful changes have been incorporated according to suggestions of worthy reviewer at line no 143-146.

Comment 41: Line 134 “…as nmol α-ketobutyrate mg protein-1 hr -1 using a spectrophotometer at 540 nm wavelength [35].

Response: Needful changes have been incorporated according to suggestions of worthy reviewer at line no 147-149.

Comment 42: Line 135 “…were characterized by streaking them on ATCC medium no.14; plates were incubated for 3 days at 111 0C (????? Check the temperature, probably it was 28ºC). The bacterial colonies showing slimes around them were characterized as EPS producing strains [35]. NOTE: Here again, check the references.

Response: Needful changes have been incorporated according to suggestions of worthy reviewer at line no 149-150.

Comment 43: Line 140 “…Agriculture Research Centre, Islamabad to evaluate the effect of isolates, 24 Rhizobium (RZ-01 to RZ-24) and 48 Plant Growth Promoting Rhizobacteria on growth attributes of chickpea seedling. Insert here the sentence:  The chosen PGPRs were isolated from rhizoplane soil (RP-01 to RP-24) and from rhizospheric soil (RS-01 to RS-24) so to screen out the 2 best strains from each category.” NOTE: Following this sentence and considering the Figure 5 and Figure 6 (see line 270 below), a sentence should be included to describe the PSB character of RP-01 to RP-24 strains.

Response: Needful changes have been incorporated according to suggestions of worthy reviewer at line no 155-158.

Comment 44: Line 144 “…isolates were sown. Uninoculated seeds were sown in the jar..”

Response: Needful changes have been incorporated according to suggestions of worthy reviewer at line no 161-162.

Comment 45: Line 145 “..control treatment. These jars were kept in the growth chamber in a complete randomized design.

Response: Needful changes have been incorporated according to suggestions of worthy reviewer at line no 162.

Comment 46: Line 147 “Among the isolates, two best Rhizobium and four PGPRs were chosen on the basis …”

Response: Needful changes have been incorporated according to suggestions of worthy reviewer at line no 163-164.

Comment 47: Line 148 “…on growth attributes of chickpea. The chosen isolates were molecularly characterized and identified..”

Response: Needful changes have been incorporated according to suggestions of worthy reviewer at line no 165-166.

Comment 48: Line 149 “…consortium effects on the productivity of chickpea at different moisture regimes.”

Response: Needful changes have been incorporated according to suggestions of worthy reviewer at line no 166.

Comment 49: Line 154 “Purified colonies of the most efficient bacterial strains were plucked and mixed with 20μL Tris-EDTA buffer in Polymerase Chain Reaction (PCR) strips.

Response: Needful changes have been incorporated according to suggestions of worthy reviewer at line no 171-172.

Comment 50: Line 156 NOTE: Not all laboratories have a PCR apparatus allowing centrifugation. Thus, the name of the apparatus should be given. The description of the PCR preparation and procedure is not clear and must be improved:  “This mixture was centrifuged in a PCR apparatus (give name) for 10 minutes at 95Co (check temperature of centrifugation,  10 min and 95ºC are usually the conditions for DNA denaturation, following centrifugation. ) and the supernatant was collected as DNA template. DNA amplification was carried out by using 2μL of forward and reverse universal primers 9F (5; 157 - GAGTTGATCCTGGCTCAG-3;) and 1510R (5;-GGCTACCTTGTTACGA-3; 158 )respectively, 25μL TAKARA Pre-mix Ex-Taq, 20μL PCR water and 1μL of DNA template. The amplified PCR products were sent to Macrogen, Koreafor sequencing and strains were identified using the EzBioCloud server.”

Response: Needful changes have been incorporated according to suggestions of worthy reviewer at line no 173-179.

Comment 51: Line 165 “…and yield attributes of chickpea in comparison to the untreated control. For the experiment…”

Response: Needful changes have been incorporated according to suggestions of worthy reviewer at line no 195.

Comment 52: Line 169 “…of formula in [37,38] and the moisture was maintained with the help of a Time Domain Reflectometer (TDR) used during the experiment. ..”~

Response: Needful changes have been incorporated according to suggestions of worthy reviewer at line no 199-200.

Comment 53: Line 171 NOTE: After the sentence “.. (both inoculated and uninoculated) were sown in respective pots.”, it should be mentioned if the experiment follows with all the germinated seeds, or if only one plant per pot is maintained.

Response: Needful changes have been incorporated according to suggestions of worthy reviewer at line no 202.

Comment 54: Line 174: NOTE: “…for their efficacy to improve productivity of chickpea with the following set of treatments:”

Response: Needful changes have been incorporated according to suggestions of worthy reviewer at line no 213.

Comment 55: Line 175, footnote of Table, remove “Where RZ-11 (Mesorhizobium ciceri) = collected from research farm of Arid Zone Research Institute, Bhakker and RZ-22 (Mesorhizobium ciceri) = collected from Chobara, district Layyah.”, because this is exposed further in Table 2.

Response: Needful changes have been incorporated according to suggestions of worthy reviewer at line no 214-215.

Comment 56: Line 181 “..in table 1. The post-harvest soil properties (Total Nitrogen and Extractable phosphorus) were determined by adopting the methods described in [42] and [43], respectively.

Response: Needful changes have been incorporated according to suggestions of worthy reviewer at line no 238.

Comment 57: Line 184, title of table “Table 1. Pre-sowing analysis of soil used in the in vitro experiment.”

Response: Needful changes have been incorporated according to suggestions of worthy reviewer at line no 240.

Comment 58: Line 186 “Proline contents in 130 days old leaves were determined following the method described in [45].

Response: Needful changes have been incorporated according to suggestions of worthy reviewer at line no 242-243.

Comment 59: Line 189 “..upper pink layer was selected for quantification using a spectrophotometer at 520 nm wavelength.”

Response: Needful changes have been incorporated according to suggestions of worthy reviewer at line no 245.

Comment 60: Line 190 “Seed protein contents were analyzed following the method described in [42].”

Response: Needful changes have been incorporated according to suggestions of worthy reviewer at line no 246.

In Results:

Comment 61: Line 197 NOTE. Section 3.1. should be removed and the information passed to “Materials and methods” section. After correcting line 199 below as suggested, move the corrected sentence to section 2.2., as explained above for line 107.

Response: Needful changes have been incorporated according to suggestions of worthy reviewer at line no 254-258.

Comment 62: Line 199 “Among them, 72 were repeatedly streaked on respective mediums in petri plates to obtain pure colonies for a screening process in growth chamber.” i.e., remove “different bacterial strains” and “finalized for the screening process”.

Response: Needful changes have been incorporated according to suggestions of worthy reviewer at line no 254-258 and corrected sentence shifted to 117-119.

Comment 63: Line 203 “Initially, 72 strains were tested for screening and among them, the 6 most promising strains were observed by Phase Contrast Microscope (Phase contrast 2, Nikon, Japan) for the colony morphology. MIRA3, Tescan SEM (scanning electron microscope) was used to characterize (insert here which characteristics) (Table 2).” NOTE: the last 5 descriptions in table 2 refer to colony morphology, so it is relevant to mention which of the characters were analyzed with MIRA.

Response: Needful changes have been incorporated according to suggestions of worthy reviewer at line no 260-263.

Comment 64: Line 206 Remove the paragraph “The only RP-01colony showed umbonate elevation with yellow color and marked as gram +ve 207 instead of all other strains that were morphologically gram –ve. The strain RP-08 (Enterobacter cloacae) …. very sticky with raised elevation and off white in color”, since the characteristics are shown in Table 2. NOTE: There are incongruencies between the text and the table, for instance, “The only RP-01colony showed umbonate elevation with yellow color” is noted “white” in the table, please check data in the table.

Response: Needful changes have been incorporated according to suggestions of worthy reviewer at line no 262-269.

Comment 65: Line 211, Table 2 For RS-14: Opaque is not an elevation character, please correct. Also, for RZ-11 and RZ-22, remove the word “off”.

Response: Needful changes have been incorporated according to suggestions of worthy reviewer at line no 269.

Comment 66: Line 213 “Identified bacterial strains were biochemically characterized. Two strains (RZ-11 and RZ214 22) showed positive results for ammonia production, whereas others remained…”

Response: Needful changes have been incorporated according to suggestions of worthy reviewer at line no 271.

Comment 67: Line 215 “..of yellowish colored zones around the colony indicated siderophore production.”

Response: Needful changes have been incorporated according to suggestions of worthy reviewer at line no 273-274.

Comment 68: Line 216 NOTE: Again, there is an incongruence with the table, where several isolates tested positive for siderophore production, mostly RP-01 and RZ-15. Again here, check your data.

Response: Needful changes have been incorporated according to suggestions of worthy reviewer at line no 275.

Comment 69: Line 217 “Regarding the HCN test, all strains tested negative. The appearance…”

Response: Needful changes have been incorporated according to suggestions of worthy reviewer at line no 275-276.

Comment 70: Line 220 “…and weak positive, respectively, for IAA production. However, for solubilization of tri-calcium phosphate, 2 strains (RP-01…”

Response: Needful changes have been incorporated according to suggestions of worthy reviewer at line no 278-280.

Comment 71: Line 222 “…solubilization index % (3.00 and 2.90), whereas RS-14 and RS-15 did not show P223 solubilization activity.”

Response: Needful changes have been incorporated according to suggestions of worthy reviewer at line no 280-281.

Comment 72: Line 223 NOTE: Again, check your data,  according to the table, only RP-08 and RS-14 had both catalase and protease activity, so the sentence should be : “Moreover, 4 PGPR strains (RS-14, RS-15, RP-01, RP-08) revealed strong potential for the catalase and/or protease enzymes production and 2 strains RS-14 and RS-15 showed positive response for amylase production (Table 3). “

Response: Needful changes have been incorporated according to suggestions of worthy reviewer at line no 283-285.

Comment 73: Line 226, Table 3 title “Characterization (Biochemical parameters) of most…”

Response: Needful changes have been incorporated according to suggestions of worthy reviewer at line no 286.

Comment 74: Line 227: Remove from the table footnote “Biochemical parameters”

Response: Needful changes have been incorporated according to suggestions of worthy reviewer at line no 288.

Comment 75: Line 231 “The identified nucleotides sequences were arranged in the phylogenetic tree (Fig. 1) using MEGA 7 [47]. Isolate RZ-11 matches to Mesorhizobium ciceri strains RZ-22, SS1 (5) and CM-25”. Delete “by 64%, 94% and 100% respectively.”

Response: Needful changes have been incorporated according to suggestions of worthy reviewer at line no 292-294.

Comment 76: Line 234 “Similarlystrain RS-14 hamatching criteria to bacteria Bacillus mojavensis strain RS-1, PMCC-9 and 235 LMB3G43 in the phylogenetic tree and RS-15 matches 100% to Providencia vermicola and is clustered to 3 strains Mum1, 236 Ag1 and OF6. “

Response: Needful changes have been incorporated according to suggestions of worthy reviewer at line no 295-296.

Comment 77: Line 236:  “PGPR isolates having P-solubilizing ability coded RP-01 and RP-08 match to Bacillus subtilis strain XGL205 and Enterobacter cloacae strain MSKrespectively.”

Response: Needful changes have been incorporated according to suggestions of worthy reviewer at line no 297-299.

Comment 78: Line 239, Figure 1 legend. “Phylogenetic tree showing the screened isolates given in bold letters and closest homologues.” Add in the figure legend what represent the numbers in the nodes. NOTE: If the numbers in the nodes are bootstrap support values, they translate the support of the monophyletic clades, therefore the proposed changes in line 231 and next lines.

Response: Needful changes have been incorporated according to suggestions of worthy reviewer at line no 301-302.

Comment 79: Line 243 “…maximum quantity of ACCD, EPS and IAA, which production averaged 0.84 μmol mg -1 protein hr-1), 0.80 mg mL-1 and 86 μmol mL-1, respectively.” Check the units, μM/ml is not possible…Check also the symbols of SI nomenclature, e.g. L is for liter.

Response: Needful changes have been incorporated according to suggestions of worthy reviewer at line no 304-309.

Comment 80: Line 244 “Isolate RP-01 showed the highest quantity (14.2 ug L-1) of P-solubilization ..”

Response: Needful changes have been incorporated according to suggestions of worthy reviewer at line no 310.

Comment 81: Line 246, Table 4.  Check the units in the table. Explain ND in a table footnote.

Response: Needful changes have been incorporated according to suggestions of worthy reviewer at line no 314.

Comment 82: Line 251 “Seventy two Isolated strains were evaluated in a trial conducted in growth chambeto screen out most efficient rhizobium and PGPR strains for different attributes of seedling growth (Fig. 3).

Response: Needful changes have been incorporated according to suggestions of worthy reviewer at line no 321-322.

Comment 83: Line 254 “..nodules per plant, respectively (Fig. 4). Among all tested isolates, two PSB strains (RP-01 and RP- 08) significantly enhanced shoot length by 19.4 cm and 19.5 cm, respectively (Fig. 5). A similar trend was recorded ..”

Response: Needful changes have been incorporated according to suggestions of worthy reviewer at line no 324.

Comment 84: Line 257 “Two PGPR isolates (RS-14 and RS-15) were found most efficient for growth and development of the plant shoot and root in screening trials. Delete “Due to strong enzymatic activities”, as this should be part of the discussion section.

Response: Needful changes have been incorporated according to suggestions of worthy reviewer at line no 326-327.

Comment 85: Line 260 “On the other hand, the highest root length (19.033 cm) was recorded for the treatment where chickpea seeds were inoculated with RS-15 followed by RS-14 (17.667 cm).”

Response: Needful changes have been incorporated according to suggestions of worthy reviewer at line no 330-332.

Comment 86: Line 264, Figure 3 legend. “Effect of strains on shoot length of chickpea in growth chamber, where..”

Response: Needful changes have been incorporated according to suggestions of worthy reviewer at line no 334-335.

Comment 87: Line 267, Figure 4 legend. “Effect of Rhizobium strains on nodulation plant-1. The bars indicate standard error (±SE) of the mean (n = 3). “NOTE: See also line 372 below.

Response: Needful changes have been incorporated according to suggestions of worthy reviewer at line no 337-339.

Comment 88: Line 270, Figure 5 legend. “Effect of PSB strains on chickpea shoot length (cm) The bars indicate standard error (±SE) of the mean (n = 3). NOTE: See also line 372 below. Also, In Figure 5 (and Figure 6), PSB strains are showed in the x-axis; this suggests that RP-01 to 24 strains are all PSB and the remained are classed as other PGPR. This is not explained in the Materials and methods section and this should be explained after the sentences suggested for line 140 above.

Response: Needful changes have been incorporated according to suggestions of worthy reviewer at line no 341-343.

Comment 89: Line 273, Figure 6 legend. “Effect of PSB strains on chickpea root length (cm). The bars indicate standard error (±SE) of the mean (n = 3). All means are significantly …” NOTE: See also line 372 below.

Response: Needful changes have been incorporated according to suggestions of worthy reviewer at line no 345-347.

Comment 90: Line 276, Figure 7. “Effect of PGPR strains on chickpea shoot length (cm). The bars indicate standard error (±SE) of the mean (n = 3). All means are significantly…” NOTE: See also line 372 below.

Response: Needful changes have been incorporated according to suggestions of worthy reviewer at line no 349-351.

Comment 91: Line 279, Figure 8. “Effect of PGPR strains on chickpea root length (cm). The bars indicate standard error (±SE) of the mean (n = 3). “NOTE: See also line 372 below.

Response: Needful changes have been incorporated according to suggestions of worthy reviewer at line no 353-355.

Comment 92: Line 284 “height (32.37 cm), number of pods plant-1 (20.67)” NOTE: pods plant-1 in the text, however per pot in the table. Thus, the correction proposed above line 171.

Response: Needful changes have been incorporated according to suggestions of worthy reviewer at line no 360.

Comment 93: Line 287 “..level 1 (55% of soil field capacity) (Table 5).”

Response: Needful changes have been incorporated according to suggestions of worthy reviewer at line no 363.

Comment 94: Line 293, Table 5. “Effect of inoculants on growth attributes of chickpea at different moisture regimes” NOTE: Check congruence with text. Also correct “No. pods per pot t-1 “to “No. pods pot-1

Response: Needful changes have been incorporated according to suggestions of worthy reviewer at line no 369.

Comment 95:  “…RZ22= Mesorhizobium ciceri, Moisture level 1= 55% of Field Capacity, Moisture level 2= 75% of Field Capacity, Moisture level 3= 95% of Field Capacity.”

Response: Needful changes have been incorporated according to suggestions of worthy reviewer at line no 373-375.

Comment 96: Line 303 “…mgg-1 DW) at same moisture levelwhile minimum proline contents (1.1533 mg g-1 DW)..NOTE: Again, check your data, as text description do not match the table..

Response: Needful changes have been incorporated according to suggestions of worthy reviewer at line no 379-380.

Comment 97: Line 304 “determined from untreated plants at moisture level 3(95% FC). Maximum grain N and protein contents…”

Response: Needful changes have been incorporated according to suggestions of worthy reviewer at line no 380.

Comment 98: Line 306 “respectively (Table 6). T2 responded most effectively at moisture level 1 (55% FC) both for grain N and protein (4.19% and 26.17%) contents. Lower N and protein contents were observed in grains of untreated plants at all given moisture levels. NOTE: Again, check your data, as text description do not match the table.

Response: Needful changes have been incorporated according to suggestions of worthy reviewer at line no 382-284.

Comment 99: Line 310 “…moisture level 2 by showing 0.35% P contents in grain followed by T4 at moisture level 3. T2 was the best performing consortium among all treatments, including the control, at moisture level 1which resulted in 0.31 % P in grains of chickpea.

Response: Needful changes have been incorporated according to suggestions of worthy reviewer at line no 387-388.

Comment 100: Line 312 “..0.31 % P in grains of chickpea. The data regarding nitrogen acquisition in chickpea shoot shows that maximum nitrogen contents (1.66% - 1.65%) were recorded for treatments T2 and T4 at moisture level 1 (55% of field capacity). Similarly, T7 appeared as the most efficient consortium with 1.65% N content in chickpea shoot at moisture level 2. Contrarily, T9 (control) showed the lowest N contents (1.31% -1.32%) in shoot of chickpea at all moisture levels.

Response: Needful changes have been incorporated according to suggestions of worthy reviewer at line no 389-393.

Comment 101: Line 317 “..the P contents in chickpea shoot (Table 6) showed the highest P (0.34%)  for T4 at moisture  level 3, followed by T2, which showed 0.32% P at the same moisture level.”

Response: Needful changes have been incorporated according to suggestions of worthy reviewer at line no 394-396.

Comment 102: Line 319. “..at moisture level 2. T9 (control) was recorded as the lowest performing treatment, which showed 0.20 - 0.22% P contents in chickpea shoots at all given moisture regimes.

Response: Needful changes have been incorporated according to suggestions of worthy reviewer at line no 396-397.

Comment 103:  “Effect of inoculants on nutrient acquisition of chickpea at different moisture regimes.”

Response: Needful changes have been incorporated according to suggestions of worthy reviewer at line no 398.

Comment 104: Line 323 “The data regarding the nitrogen percentage in post-harvest soil (Fig.9) show that the highest  N (0.016%) was recorded for the treatments T4, T5 and T7 at moisture level 3followed by T2 with  0.057% (NOTE: It cannot be this value, it has to be lower than 0.016%, check your data) N in post-harvest soil at moisture level 1. “Similarly, T7 gave 0.057% N at moisture level 1 and 2. T9 (control) was recorded as the lowest performing treatmentwhich showed 0.012% N in post-harvest soil, at all given moisture levels.

Response: Needful changes have been incorporated according to suggestions of worthy reviewer at line no 415-418.

Comment 105: Line 326 “..level 1 and 2. T9 (control) was recorded as the lowest performing treatmentwhich showed  0.012% N in post-harvest soil at all given moisture levels.

Response: Needful changes have been incorporated according to suggestions of worthy reviewer at line no 419.

Comment 106: Line 328 “…was recorded for both treatments T2 and T7 at moisture level 1 (55% of field capacity).

Response: Needful changes have been incorporated according to suggestions of worthy reviewer at line no 421-422.

Comment 107: Line 329 “…(55% of field capacity). T7 was recorded as the most efficient consortium by showing 4.166 ppm post-harvest soil P at moisture level 3. T9 (control) showed the lowest P contents (3.00 ppm NOTE: here again, the number was incorrect, please check your data, figures and text have to match…) in post-harvest soil at all moisture levels.

Response: Needful changes have been incorporated according to suggestions of worthy reviewer at line no 422-424.

Comment 108: Line 333, Figure 9. NOTE (the same note applies to Figure 10): Letters should be included over the bars to represent the significantly different means.

Response: Needful changes have been incorporated according to suggestions of worthy reviewer at line no at line no 425.

Comment 109: Line 335 “The bars indicate standard error (±SE) of the mean (n = 3

Response: Needful changes have been incorporated according to suggestions of worthy reviewer at line no 430.

Comment 110: Line 337 “…T9=control, where RZ11= Mesorhizobium ciceri, …”

Response: Needful changes have been incorporated according to suggestions of worthy reviewer at line no 433-435.

Comment 111: Line 339 “…Moisture level 1= 55% of Field Capacity, Moisture level 2= 75% of Field Capacity, Moisture levels 3= 95% of Field Capacity”

 Response: Needful changes have been incorporated according to suggestions of worthy reviewer at line no 440-445.

In Discussion:

Comment 112: Line 355 “..by [48], who showed that some soil microbes produce ammonia and enhanced crop growth attributes.

Response: Needful changes have been incorporated according to suggestions of worthy reviewer at line no 451.

Comment 113: Line 355 “Two PGPR isolates, RS-14 (Bacillus mojavensis) and RS-15 (Providencia vermicola), were evaluated as IAA producing bacterial strains.”

Response: Needful changes have been incorporated according to suggestions of worthy reviewer at line no 452.

Comment 114: Line 357 “Several microbes produce active auxin as IAA, which is a plant growth promoter [49,50]. “

Response: Needful changes have been incorporated according to suggestions of worthy reviewer at line no 452-454.

Comment 115: Line 358 “…”Two PGPR isolates RP-01 (Bacillus subtilis) and RP-08 (Enterobacter cloacaewere evaluated as most …”

Response: Needful changes have been incorporated according to suggestions of worthy reviewer at line no 454-455.

Comment 116: Line 359 “..phosphate solubilizers. The results are also in agreement with the findings of [51], who observed the maximum phosphate solubilizing ability for Enterobacter sp.”

Response: Needful changes have been incorporated according to suggestions of worthy reviewer at line no 455-456.

Comment 117: Line 361 “..zones surrounding the microbial colonies could be due to the synthesis of organic acids with low molecular weight, or due to polysaccharides and phosphatase production by phosphate solubilizing microbes 363 [52].”

Response: Needful changes have been incorporated according to suggestions of worthy reviewer at line no 457-458.

Comment 118: Line 368 (to 371) “We selected 24 bacterial strains each ….” Delete, as it was passed to section 2.5. (see line 140).

Response: Needful changes have been incorporated according to suggestions of worthy reviewer at line no 465-468.

Comment 119: Line 372 “..Nodulation plant-1 372 was increased 100% by rhizobium strains (RZ-11, RZ-22) as compared …” NOTE: If 25 represents the control in Figure 4, nodulation has increased ca tenfold as compared to uninoculated plants. Also, Figures 4 to Figure 8 should explain in the legend that 25 corresponds to the non-inoculated plants.

Response: Needful changes have been incorporated according to suggestions of worthy reviewer at line no 470-471.

Comment 120: Line 374 “..reported that Mesorhizobium ciceri inoculation increased significantly the nodulation of chickpea plants..”

Response: Needful changes have been incorporated according to suggestions of worthy reviewer at line no 471.

Comment 121: Line378 “..length by 54.4 % and 54.9 % over the un-inoculated control.”

Response: Needful changes have been incorporated according to suggestions of worthy reviewer at line no 475.

Comment 122: Line 380 “The root and shoot length may be increased due to increase availability of phosphorus thanks to PSBs activity, since phosphorus has an important role in root development and cell division. All plant growth promoting isolates showed significant improvement in seedling shoot length, between 6.39% and 45.82%, as compared to the control; two PGPR isolates, RS-14 and RS-15having phytohormonal activityincreased chickpea shoot length 42.73% and 45.82% ,respectively, in comparison to the untreated control.

Response: Needful changes have been incorporated according to suggestions of worthy reviewer at line no 477-481.

Comment 123: Line 385 “Moreover, PGPR strains (RS-01 to RS-24) showed a significantly positive response in chickpea seedlings root length (from 0.55% to 56.87%) as compared to the control.”

Response: Needful changes have been incorporated according to suggestions of worthy reviewer at line no 482-484.

Comment 124: Line 388 “..and RS-14, respectively, over the control without inoculation.”

Response: Needful changes have been incorporated according to suggestions of worthy reviewer at line no 486.

Comment 125: Line 390 “Our experimental results are also supported by findings of (Insert here name of the authors )[60], who showed that plant-root system increases up to 40% in PGPRs treated plants as compared to untreated controls.”

Response: Needful changes have been incorporated according to suggestions of worthy reviewer at line no 489.

Comment 126: Line 393 “Six most efficient strains were used to make 8 different prolific consortia that were evaluated..”

Response: Needful changes have been incorporated according to suggestions of worthy reviewer at line no 491.

Comment 127: Line 395 “T2 consortium (Mesorhizobium ciceri RZ-11 + Bacillus subtilis RP-01+ Bacillus mojavensis RS-14) was found most effective for chickpea growth and yield attributes at moisture level 1which was maintained on 55 % of field capacity.”

Response: Needful changes have been incorporated according to suggestions of worthy reviewer at line no 493-495.

Comment 128: Line 398 “..19.8% in root length, 29% in 100 grain weight, 46.9% in  biological yield pot-1, 50% in economic yield pot-1  compared to the untreated control at moisture level 1 (55 % of FC).”

Response: Needful changes have been incorporated according to suggestions of worthy reviewer at line no 497-498.

Comment 129: Line 402 “..production by PGPRs from the consortia in the applied treatment.”

Response: Needful changes have been incorporated according to suggestions of worthy reviewer at line no 500-501.

Comment 130: Line 405 “..with the un-inoculated treatment. Quantity of nodules, economic yield and biological yield were also increased by 73.53%, 31.76% and 24.37% respectively, over the control.”

Response: Needful changes have been incorporated according to suggestions of worthy reviewer at line no 503-505.

Comment 131: Line 408 “..at moisture level 1 and 3 (55 % & 95 % FC)respectively, over the control. T2 resulted in postharvest soil N and P increases of 30% and 50%, respectively, in comparison to the un-inoculated treatment at 55 % of field capacity.”

Response: Needful changes have been incorporated according to suggestions of worthy reviewer at line no 507-509.

Comment 132: Line 410 “Also in the present study, the calculated proline content of chickpea leaves showed an increasing trend at moisture level 1 (55 % of FC) by inoculation with T2-consortium having ACCD activityand EPS and IAA production.”

Response: Needful changes have been incorporated according to suggestions of worthy reviewer at line no 509-512.

Comment 133: Line 413 “..[62]. Statistically higher grain yield (32.76%, Please, check this valueaccording to the table is 35.50) was obtained with T7 treatment (Mesorhizobium ciceri RZ-22 + Enterobacter Cloacae RP-08 + Providencia vermicola RS-15) at moisture level 2 (75% of FC) and T4 (Mesorhizobium ciceri RZ-11 + Enterobacter Cloacae RP-08 + Providencia vermicola RS-15) (31%, Please, check this valueaccording to the table 27.53%) at moisture level 3 (95% of FC), as compared to untreated chickpea plants. NOTE: several times in the manuscript Enterobacter is misspelled!

Response: Needful changes have been incorporated according to suggestions of worthy reviewer at line no 513-519.

Comment 134: Line 417 “Hence, microbial combinations in T4 and T7 could be used to make effective biofertilizers for chickpea growing areas under rainfed conditions to help plants to cope with

drought spells.”

Response: Needful changes have been incorporated according to suggestions of worthy reviewer at line no 519-521.

Comment 135: Line 422 “…of drought through experimentation on desert soil is quite a novel idea being adopted in the present study.” NOTE: it is a valid idea, but not so novel…

Response: Needful changes have been incorporated according to suggestions of worthy reviewer at line no 525.

Comment 136: Line 425 “..the vital role of isolated strains to be utilized as bio-fertilizers under drought spell in Thal desert, a main chickpea producing area in Pakistanwhich is the 4th  largest chickpea producing country.

Response: Needful changes have been incorporated according to suggestions of worthy reviewer at line no 528-529.

Comment 137: Line 426 “..producing country. In this study, we found that the consortium T2 (Mesorhizobium ciceri RZ-11 + Bacillus subtilis RP-01+Bacillus mojavensis 428 RS-14) can perform best in drought conditions (55% field capacity)..”

 Response: Needful changes have been incorporated according to suggestions of worthy reviewer at line no 530.

In references:

Comment 138: NOTE: Check your references, those must be in agreement with the text, for instance see line 116 above.

Response: Needful changes have been incorporated according to suggestions of worthy reviewer at line no 579-695.

Reviewer 2 Report

This study investigated the effect of co-inoculated bacterial strains on growth of chickpea under different moisture regimes.

I appreciated that the strains were collected, isolated, characterized, and screened. I appreciated that yield was included in the study. I think this information will be of interest to readers, and provide opportunities for further study in this area.

There are minor/moderate spelling and grammar edits needed throughout article.

I would have liked to see more than 3 replications, but I am assuming this number is justified.

Author Response

This study investigated the effect of co-inoculated bacterial strains on growth of chickpea under different moisture regimes.

I appreciated that the strains were collected, isolated, characterized, and screened. I appreciated that yield was included in the study. I think this information will be of interest to readers, and provide opportunities for further study in this area.

Comment 1: There are minor/moderate spelling and grammar edits needed throughout article.

Response: As per recommendation, the whole manuscript is revised and improved for linguistic and technical mistakes carefully.

I would have liked to see more than 3 replications, but I am assuming this number is justified.

Round 2

Reviewer 1 Report

The manuscript entitled “Desert Soil Microbes as a Mineral Nutrient Acquisition tool for Chickpea (Cicer arietinum L.) Productivity at Different Moisture Regimes” by Aulakh et al. describes the isolation and characterization of nitrogen fixing rhizobia and PGPR from desert soil and the further establishment of consortia, which were analyzed in terms of growth and development of chickpea in desert soil. The latter version of the manuscript has greatly improved. Still, further minor modifications are proposed below. Note that they include also further changes on previously modified sentences.

I advise the authors to make a final revision of the manuscript after introducing all changes, as I still noticed some incongruences, for instance SI units and axis titles. Also, a final English revision by a native English speaker is highly recommended.

In introduction:

Lines 64-65.Due to phosphorus availability, the surface area, volume and root length increases, thus phosphorus indirectly promotes water absorption from untapped soil profile.”

Line73:”…quantity of phosphorus but in unavailable form [19].”

In Materials and methods:

Line 115 “..respectively. The colonies were incubated at 28 ± 2ªC for 24 to 48 hours.” NOTE: I would delete PGPR, since you test if the colonies are PGPR after isolation.

Line 146 “Afterwards, the plates with the bacterial strains were incubated for 3 days at 28ºC and checked for colony growth [34]. Delete “showing the ACC deaminase activity”.

Line 148 “..quantified as nmol α-ketobutyrate mg protein-1 h-1

Line 173-174 NOTE: I did not find that the referred thermal cycler had an incorporated function for centrifugation. I believe thus, the correct sentence would be: “This mixture was placed in a PCR apparatus (Thermal Cycler PCR PEQSTAR, Germany) for 10 minutes at 95ºC to extract the template DNA, which was collected in the supernatant after centrifugation. DNA amplification was carried in the same apparatus using 2µL..”

Lines 183-185 “..to test their compatibility. For that purpose, isolates were refreshed overnight on 25 ml of Nutrient broth (NB) medium and inocula of 2 µL, each containing 10 6 bacterial cells of a distinct isolate were inoculated 1 cm apart on NA medium in one petri plate. NOTE: Please use NA for agar based-media and NB for liquid media (see also line 206 below).

Line 190 “..consortia were quantified using the methods as discussed in section (2.4).”  Delete “Biochemical Characterization of isolates.”

Line 202 “..respective pots and the experiment was followed with all the germinated seeds. “

Line 206 “…individually in 50 ml NB and further incubated overnight at 30 ºC, 150 rpm. NOTE: Please delete “agar”.

Line 210-211 “…optical density was measured at 550nm and adjusted to a concentration of”. Delete spectrophotometer and wavelength.

Line 212 “Strain suspensions were combined in equal amounts (i.e. equal cfu mL-1) to prepare the respective consortium for each treatment”, instead ofResultant suspension of each strain…”

Line 213 “…for their efficacy to improve chickpea productivity with the following set of treatments:”

Line 216 “2.8.1. Seed inoculation with consortia”

Line 217 -218 “The chickpea seeds were washed with sterile water and rinsed with 70 % ethanol. Seeds were then immersed in 6.5 % sodium hypochlorite …”

Line 220-221 “The seeds for the untreated control were soaked in sterile water (Molina Romero--- et al.) [40].

Line 223-224 “…of the chickpea plants 60 days after seed inoculation through plate count method. 

Lines 224-229 “through plate count method. Calculations were performed on the basis of serial 10 folds dilution in duplicate, using the pour plate method, replicated samples of 1g soil and an appropriate dilution (Johnson and Curl, 1972)[41]. Each value is presented as an average of three individual plate counts of the colonies of the PGPR isolates within a consortium (NOTE: is this what was made? see lines 400-404 lines below). Petri dishes (90mm diameter) contained 25ml of (NOTE: Name of the medium here) medium and plates were incubated at 28-30 ºC. Colony forming units (CFU) were recorded after 48 hours, the average number per gram oven dry weight of soil was calculated as

Line 232 Delete the sentence “Where, CFU is Colony Forming Unit”

Line 234 “Three composite samples were collected from the pile of pre-sowing soil for physico-chemical analysis; texture [42], organic matter [43] total phosphorus [44], total nitrogen [45], extractable phosphorus and extractable potassium [46], and soil pH (1:5 soil–water) were determined following the methodology described by [47].”

Line 240 “Table 1. Pre-sowing analysis of soil used in the glass house experiment.” NOTE: considering the definition, there is no in vitro experiment in the work.

Line 242 “Proline contents in old leaves were determined following the method”. Delete the word “by”.

Line 246 “Seed protein contents were analyzed following the method…”

Line 250 “by adopting a complete Randomized Design (CRD).”

In Results:

Lines 277 “Among the isolates, 2 strains (RP08 and RS-15) were found strong positive, while RP-01 and RS-14) were observed as moderate IAA producers and Rhizobium isolates tested negative.”

Line 296 “..100% to Providencia vermicola  and is clustered together with 3 Providencia strains Mum1, Ag1 and OF6.”

Line 302 “Bootstrap values (n = 100) are displayed at the nodes.”

Line 306: “..However, the strain RP-08 showed maximum values of ACCD, EPS and IAA, which averaged 0.84 (μM/mg protein/h), 0.80 (mg/mL) and 86 (μg/mL), respectively. Table 4 showed that T2 consortium was found efficient for ACCD and IAA (2.6 μM/mg protein/h and 177 μg/mL) respectively, followed by T7 with ACCD (2.5 μM/mg protein/h) and IAA (171 μg/mL). Isolate RP01 showed the highest value (14.2 ugLl-1) of P-solubilization compared to other strains”. NOTE: Please, be congruent in the presentation of units, for instance mg/ mL or mg mL-1 throughout the manuscript and particularly in table 4. If you use mg/ mL, then use ug /L for Phosphate solubilization. Check for the preferences in unit presentation in the guide for authors section of the journal.

Line 312 “Table 4. Quantitative estimation of EPS and IAA production, and ACC-deaminase and Phosphate solubilization activities.

Line 338: “Strains (1-24) are rhizobium (RZ-01 to RZ-24) while 25 is the control treatment. “Remove “depicted in the graph”. NOTE. According to the materials and methods section, the control corresponds to uninoculated seeds, so 25 should not be a strain.. If a different control was applied, for instance a non-nodulating strain, explain in the legend.

Line 341. “Strains (1-24) are phosphate solubilizers (RP-01 to RP-24) while 25 is the control treatment”.

Line 349 “Strains (1-24) are PGPRs (RS-01 to RS-24) while 25 is the control treatment”

Line 353 “. Strains (1-24) are PGPRs (RS-01 to RS-24) while 25 is the control treatment “

Line 357 “The isolates to be used for making each consortium were appeared compatible to each other, thus consortia were built and inoculated to elucidate prominent…”

Line 381 “..protein contents (4.31% & 26.96%) were recorded for T4 at moisture level 3 (95% FC)..

Line 382 “..4.28% and 26.79% with application of T7 at moisture level 2 (75% FC), respectively (Table 6).”

Line 385 “…Similarly, T7 was the most promising consortium on moisture level 2, showing 0.35% P contents in grain followed by T4 at moisture level 3. T2 was the best performing consortium among all treatments, including the control, at moisture level 1…”

Line 389 “..chick-pea shoot shows that the maximum nitrogen contents (1.66% - 1.65%) were recorded for treatments T2 and T4 at moisture level 1 (55% of field capacity).”

Line 391 “..as the most efficient consortium with 1.65% N content..”

Line 400-404 “The data regarding the isolate population in rhizospheric soil (Fig.9) shows that the maximum Colony Forming Unit (2.45^108) value was recorded for the treatment T2 at moisture level 3 followed by T7 with 2.36^108 at the same moisture level. T2 and T7 showed 1.67^108 and 1.36^108, respectively, at moisture level 1. T9 (control) showed zero population of the tested consortia isolates at all given moisture levels.

NOTE: What are these values? The total number of PGPR colonies that compose the consortium? For example, Bacillus subtilis and Providencia vermicola in T1? This must be explained, because it is understood that the watering was not performed with sterile water, so it is difficult to get a zero bacterial population for the control, so I believe you refer to the  absence of any colony of the PGPR isolates under study. The authors should be precise in referring what are they counting exactly, so I propose that section 2.8.2 in materials and methods explain that (see lines 224-229 above.)

Line 406 “Figure 9. Survival of isolates (CFU x 106 g-1 (?)) in rhizospheric soil 60 days after sowing (DAS) at different moisture levels in glass house experiment. NOTE: According to section 2.8.2 the cfu value herein should be given per unit of dry weight (cfu/g (?) soil). Please, check Y axis and legend of Figure 9 accordingly.

Line 422 “T7 was recorded as the most efficient consortium at moisture level 3 (4.166 ppm post-harvest soil P).”

In Discussion:

Line 454 “Two PGPR isolates RP-01 (Bacillus subtilis) and RP455 08 (Enterobacter cloacae) were IAA producers and the most efficient phosphate solubilizers.”

Line 465 “..by producing indole acetic acid (IAA) and ACC-deaminase to reduce ethylene levels in roots.”

Line 469  “In the growth chamber assay for isolates’ screening, nodulation plant -1 was increased over (Please, verify) 100%.. NOTE: If you consider the average nodules/plant value of all strains versus 25 (control) the percentage is higher than 100% (Figure 4). Please verify.

Line 477 “. The root and shoot length may be increased due to increased availability of phosphorus thanks to PSBs activity, as phosphorus has an important role in root development and cell division.”

Line 499-500 “Increments in the yield and yield attributes of chickpea might be due to ammonia production by rhizobium, and IAA production, ACC deaminase and PSB activity of PGPRS from the consortia in the applied treatment.” NOTE: Remove EPS, as this was not tested in consortia.

Line 504: “Quantity of nodules, economic yield and biological yield were also increased for (Please, insert in the sentence the name of the consortium that gives these values) by 73.53%, 31.76% and 505 24.37% respectively, over the control. “

Line 505-507 “..24.37% respectively, over the control. The physiological attributes of chickpea as grain N and protein content were statistically increased by 7.6% over the control, with application of T2 at moisture level 1 (55 % FC) and by 10.12% (grain N) with T4 at moisture  level 3 (95 % FC). NOTE: Please check the calculation, only in N grain for T4 moisture level 3, not for protein.

Line 509-512 “Also in the present study, the calculated proline content of chickpea leaves showed an increase at moisture level 1 (55 % of FC) by inoculation with T2-consortium having ACCD activity, and IAA production.” Delete EPS, since it was not tested.

Line 513 ” Statistically higher grain yield (35.49 %) over that of untreated chickpea plants was obtained with T7 treatment (Mesorhizobium ciceri RZ-22 + Enterobacter Cloacae RP-08 + Providencia vermicola RS-15) at moisture level 2 (75% of FC) and with T2 (Mesorhizobium ciceri RZ-11 + Bacillus subtilis  RP-01+ Bacillus mojavensis RS-14) (50% higher grain yield) at  moisture level 1 (55% of FC).”

Line 517 “T4 (Mesorhizobium ciceri RZ-11 + Enterobacter Cloacae RP-08 + Providencia vermicola RS-15) had 27.53% better economic yield at moisture level 3 (95% of FC) as compared to uninoculated chickpea plants.”

Line 519 “Hence, microbial combinations…”

In Conclusions:

Line 525 “..desert soil is a valid idea”. Delete the word “quite”.

Line 526 “Here, series of experiments revealed growth promotion as well as substantial nodulation in chickpea that enhanced its grain yield under drought stress.”

Line 528 “..under drought spell in Thal desert, a main chickpea producing area in..”

Author Response

Dear Editor,

Kindly find below the incorporated amendments and response being suggested by the reviewer in the manuscript which are as follows;

The manuscript entitled “Desert Soil Microbes as a Mineral Nutrient Acquisition tool for Chickpea (Cicer arietinum L.) Productivity at Different Moisture Regimes” by Aulakh et al. describes the isolation and characterization of nitrogen fixing rhizobia and PGPR from desert soil and the further establishment of consortia, which were analyzed in terms of growth and development of chickpea in desert soil. The latter version of the manuscript has greatly improved. Still, further minor modifications are proposed below. Note that they include also further changes on previously modified sentences.

I advise the authors to make a final revision of the manuscript after introducing all changes, as I still noticed some incongruences, for instance SI units and axis titles. Also, a final English revision by a native English speaker is highly recommended.

Response: The MS has been extensively edited based on the above comments.

In introduction:

Comment 1: Lines 64-65. “Due to phosphorus availability, the surface area, volume and root length increases, thus phosphorus indirectly promotes water absorption from untapped soil profile.”

Response: Needful changes have been incorporated according to suggestions of worthy reviewer at line no 65.

Comment 2: Line73:”…quantity of phosphorus but in unavailable form [19].”

Response: Needful changes have been incorporated according to suggestions of worthy reviewer at line no 73.

 In Materials and methods:

Comment 3: Line 115 “..respectively. The colonies were incubated at 28 ± 2ªC for 24 to 48 hours.” NOTE: I would delete PGPR, since you test if the colonies are PGPR after isolation.

Response: Needful changes have been incorporated according to suggestions of worthy reviewer at line no 117.

Comment 4: Line 146 “Afterwards, the plates with the bacterial strains were incubated for 3 days at 28ºC and checked for colony growth [34]. Delete “showing the ACC deaminase activity”.

Response: Needful changes have been incorporated according to suggestions of worthy reviewer at line no 148.

Comment 5: Line 148 “..quantified as nmol α-ketobutyrate mg protein-1 h-1

Response: Needful changes have been incorporated according to suggestions of worthy reviewer at line no 149.

Comment 6: Line 173-174 NOTE: I did not find that the referred thermal cycler had an incorporated function for centrifugation. I believe thus, the correct sentence would be: “This mixture was placed in a PCR apparatus (Thermal Cycler PCR PEQSTAR, Germany) for 10 minutes at 95ºC to extract the template DNA, which was collected in the supernatant after centrifugation. DNA amplification was carried in the same apparatus using 2µL..”

Response: Needful changes have been incorporated according to suggestions of worthy reviewer at line no 174-176.

Comment 7: Lines 183-185 “..to test their compatibility. For that purpose, isolates were refreshed overnight on 25 ml of Nutrient broth (NB) medium and inocula of 2 µL, each containing 10 6 bacterial cells of a distinct isolate were inoculated 1 cm apart on NA medium in one petri plate. NOTE: Please use NA for agar based-media and NB for liquid media (see also line 206 below).

Response: Needful changes have been incorporated according to suggestions of worthy reviewer at line no 185-187.

Comment 8: Line 190 “..consortia were quantified using the methods as discussed in section (2.4).”  Delete “Biochemical Characterization of isolates.”

Response: Needful changes have been incorporated according to suggestions of worthy reviewer at line no 192.

Comment 9: Line 202 “..respective pots and the experiment was followed with all the germinated seeds. “

Response: Needful changes have been incorporated according to suggestions of worthy reviewer at line no 203.

Comment 10: Line 206 “…individually in 50 ml NB and further incubated overnight at 30 ºC, 150 rpm. NOTE: Please delete “agar”.

Response: Needful changes have been incorporated according to suggestions of worthy reviewer at line no 207.

Comment 11: Line 210-211 “…optical density was measured at 550nm and adjusted to concentration of”. Delete spectrophotometer and wavelength.

Response: Needful changes have been incorporated according to suggestions of worthy reviewer at line no 210-211.

Comment 12: Line 212 “Strain suspensions were combined in equal amounts (i.e. equal cfu mL-1) to prepare the respective consortium for each treatment”, instead of “Resultant suspension of each strain…”

Response: Needful changes have been incorporated according to suggestions of worthy reviewer at line no 211-212.

Comment 13: Line 213 “…for their efficacy to improve chickpea productivity with the following set of treatments:”

Response: Needful changes have been incorporated according to suggestions of worthy reviewer at line no 213.

Comment 14: Line 216 “2.8.1. Seed inoculation with consortia”

Response: Needful changes have been incorporated according to suggestions of worthy reviewer at line no 217.

Comment 15: Line 217 -218 “The chickpea seeds were washed with sterile water and rinsed with 70 % ethanol. Seeds were then immersed in 6.5 % sodium hypochlorite …”

Response: Needful changes have been incorporated according to suggestions of worthy reviewer at line no 218.

Comment 16: Line 220-221 “The seeds for the untreated control were soaked in sterile water (Molina Romero--- et al.) [40].

Response: Needful changes have been incorporated according to suggestions of worthy reviewer at line no 221.

Comment 17: Line 223-224 “…of the chickpea plants 60 days after seed inoculation through plate count method. 

Response: Needful changes have been incorporated according to suggestions of worthy reviewer at line no 225.

Comment 18: Lines 224-229 “through plate count method. Calculations were performed on the basis of serial 10 folds dilution in duplicate, using the pour plate method, replicated samples of 1g soil and an appropriate dilution (Johnson and Curl, 1972)[41]. Each value is presented as an average of three individual plate counts of the colonies of the PGPR isolates within a consortium (NOTE: is this what was made? see lines 400-404 lines below). Petri dishes (90mm diameter) contained 25ml of (NOTE: Name of the medium here) medium and plates were incubated at 28-30 ºC. Colony forming units (CFU) were recorded after 48 hours, the average number per gram oven dry weight of soil was calculated as

Response: Needful changes have been incorporated according to suggestions of worthy reviewer at line no 224-229.

Comment 19: Line 232 Delete the sentence “Where, CFU is Colony Forming Unit”

Response: Needful changes have been incorporated according to suggestions of worthy reviewer at line no 234.

Comment 20: Line 234 “Three composite samples were collected from the pile of pre-sowing soil for physico-chemical analysis; texture [42], organic matter [43] total phosphorus [44], total nitrogen [45], extractable phosphorus and extractable potassium [46], and soil pH (1:5 soil–water) were determined following the methodology described by [47].”

Response: Needful changes have been incorporated according to suggestions of worthy reviewer at line no 237-239.

Comment 21: Line 240 “Table 1. Pre-sowing analysis of soil used in the glass house experiment.” NOTE: considering the definition, there is no in vitro experiment in the work.

Response: Needful changes have been incorporated according to suggestions of worthy reviewer at line no 243.

Comment 22: Line 242 “Proline contents in old leaves were determined following the method”. Delete the word “by”.

Response: Needful changes have been incorporated according to suggestions of worthy reviewer at line no 245.

Comment 23: Line 246 “Seed protein contents were analyzed following the method…”

Response: Needful changes have been incorporated according to suggestions of worthy reviewer at line no 249.

Comment 24: Line 250 “by adopting a complete Randomized Design (CRD).”

 Response: Needful changes have been incorporated according to suggestions of worthy reviewer at line no 253.

In Results:

Comment 25: Lines 277 “Among the isolates, 2 strains (RP08 and RS-15) were found strong positive, while RP-01 and RS-14) were observed as moderate IAA producers and Rhizobium isolates tested negative.”

Response: Needful changes have been incorporated according to suggestions of worthy reviewer at line no 280-282.

Comment 26: Line 296 “..100% to Providencia vermicola  and is clustered together with 3 Providencia strains Mum1, Ag1 and OF6.”

Response: Needful changes have been incorporated according to suggestions of worthy reviewer at line no 300.

Comment 27: Line 302 “Bootstrap values (n = 100) are displayed at the nodes.”

Response: Needful changes have been incorporated according to suggestions of worthy reviewer at line no 305.

Comment 28: Line 306: “..However, the strain RP-08 showed maximum values of ACCD, EPS and IAA, which averaged 0.84 (μM/mg protein/h), 0.80 (mg/mL) and 86 (μg/mL), respectively. Table 4 showed that T2 consortium was found efficient for ACCD and IAA (2.6 μM/mg protein/h and 177 μg/mL) respectively, followed by T7 with ACCD (2.5 μM/mg protein/h) and IAA (171 μg/mL). Isolate RP01 showed the highest value (14.2 ugLl-1) of P-solubilization compared to other strains”. NOTE: Please, be congruent in the presentation of units, for instance mg/ mL or mg mL-1 throughout the manuscript and particularly in table 4. If you use mg/ mL, then use ug /L for Phosphate solubilization. Check for the preferences in unit presentation in the guide for authors section of the journal.

Response: Needful changes have been incorporated according to suggestions of worthy reviewer at line no 309-313.

Comment 29: Line 312 “Table 4. Quantitative estimation of EPS and IAA production, and ACC-deaminase and Phosphate solubilization activities.

Response: Needful changes have been incorporated according to suggestions of worthy reviewer at line no 316-317.

Comment 30: Line 338: “Strains (1-24) are rhizobium (RZ-01 to RZ-24) while 25 is the control treatment. “Remove “depicted in the graph”. NOTE. According to the materials and methods section, the control corresponds to uninoculated seeds, so 25 should not be a strain.. If a different control was applied, for instance a non-nodulating strain, explain in the legend.

Response: Needful changes have been incorporated according to suggestions of worthy reviewer at line no 342.

Comment 31: Line 341. “Strains (1-24) are phosphate solubilizers (RP-01 to RP-24) while 25 is the control treatment”.

Response: Needful changes have been incorporated according to suggestions of worthy reviewer at line no 346.

Comment 32: Line 349 “Strains (1-24) are PGPRs (RS-01 to RS-24) while 25 is the control treatment”

Response: Needful changes have been incorporated according to suggestions of worthy reviewer at line no 353.

Comment 33: Line 353 “. Strains (1-24) are PGPRs (RS-01 to RS-24) while 25 is the control treatment “

Response: Needful changes have been incorporated according to suggestions of worthy reviewer at line no 357.

Comment 34: Line 357 “The isolates to be used for making each consortium were appeared compatible to each other, thus consortia were built and inoculated to elucidate prominent…”

Response: Needful changes have been incorporated according to suggestions of worthy reviewer at line no 361-362.

Comment 35: Line 381 “..protein contents (4.31% & 26.96%) were recorded for T4 at moisture level 3 (95% FC)..

Response: Needful changes have been incorporated according to suggestions of worthy reviewer at line no 386.

Comment 36: Line 382 “..4.28% and 26.79% with application of T7 at moisture level 2 (75% FC), respectively (Table 6).”

Response: Needful changes have been incorporated according to suggestions of worthy reviewer at line no 387.

Comment 37: Line 385 “…Similarly, T7 was the most promising consortium on moisture level 2, showing 0.35% P contents in grain followed by T4 at moisture level 3. T2 was the best performing consortium among all treatments, including the control, at moisture level 1…”

Response: Needful changes have been incorporated according to suggestions of worthy reviewer at line no 390-393.

Comment 38: Line 389 “..chick-pea shoot shows that the maximum nitrogen contents (1.66% - 1.65%) were recorded for treatments T2 and T4 at moisture level 1 (55% of field capacity).”

Response: Needful changes have been incorporated according to suggestions of worthy reviewer at line no 395.

Comment 39: Line 391 “..as the most efficient consortium with 1.65% N content..”

Response: Needful changes have been incorporated according to suggestions of worthy reviewer at line no 396.

Comment 40: Line 400-404 “The data regarding the isolate population in rhizospheric soil (Fig.9) shows that the maximum Colony Forming Unit (2.45^108value was recorded for the treatment T2 at moisture level 3 followed by T7 with 2.36^108 at the same moisture level. T2 and T7 showed 1.67^108 and 1.36^108, respectively, at moisture level 1. T9 (control) showed zero population of the tested consortia isolates at all given moisture levels.

NOTE: What are these values? The total number of PGPR colonies that compose the consortium? For example, Bacillus subtilis and Providencia vermicola in T1? This must be explained, because it is understood that the watering was not performed with sterile water, so it is difficult to get a zero bacterial population for the control, so I believe you refer to the  absence of any colony of the PGPR isolates under study. The authors should be precise in referring what are they counting exactly, so I propose that section 2.8.2 in materials and methods explain that (see lines 224-229 above.)

Response: Needful changes have been incorporated according to suggestions of worthy reviewer at line no 407-409.

Comment 41: Line 406 “Figure 9. Survival of isolates (CFU x 106 g-1 (?)) in rhizospheric soil 60 days after sowing (DAS) at different moisture levels in glass house experiment. NOTE: According to section 2.8.2 the cfu value herein should be given per unit of dry weight (cfu/g (?) soil). Please, check Y axis and legend of Figure 9 accordingly.

Response: Needful changes have been incorporated according to suggestions of worthy reviewer at line no 410-412.

Comment 42: Line 422 “T7 was recorded as the most efficient consortium at moisture level 3 (4.166 ppm post-harvest soil P).”

Response: Needful changes have been incorporated according to suggestions of worthy reviewer at line no 428-429.

In Discussion:

Comment 43: Line 454 “Two PGPR isolates RP-01 (Bacillus subtilis) and RP455 08 (Enterobacter cloacaewere IAA producers and the most efficient phosphate solubilizers.”

Response: Needful changes have been incorporated according to suggestions of worthy reviewer at line no 461.

Comment 44: Line 465 “..by producing indole acetic acid (IAA) and ACC-deaminase to reduce ethylene levels in roots.”

Response: Needful changes have been incorporated according to suggestions of worthy reviewer at line no 471-472.

Comment 45: Line 469  “In the growth chamber assay for isolates’ screening, nodulation plant -1 was increased over (Please, verify) 100%.. NOTE: If you consider the average nodules/plant value of all strains versus 25 (control) the percentage is higher than 100% (Figure 4). Please verify.

Response: Needful changes have been incorporated according to suggestions of worthy reviewer at line no 476-477.

Comment 46: Line 477 “. The root and shoot length may be increased due to increaseavailability of phosphorus thanks to PSBs activity, as phosphorus has an important role in root development and cell division.”

Response: Needful changes have been incorporated according to suggestions of worthy reviewer at line no 485.

Comment 47: Line 499-500 “Increments in the yield and yield attributes of chickpea might be due to ammonia production by rhizobium, and IAA production, ACC deaminase and PSB activity of PGPRS from the consortia in the applied treatment.” NOTE: Remove EPS, as this was not tested in consortia.

Response: Needful changes have been incorporated according to suggestions of worthy reviewer at line no 507.

Comment 48: Line 504: “Quantity of nodules, economic yield and biological yield were also increased for (Please, insert in the sentence the name of the consortium that gives these values) by 73.53%, 31.76% and 505 24.37% respectively, over the control. “

Response: Needful changes have been incorporated according to suggestions of worthy reviewer at line no 511-512.

Comment 49: Line 505-507 “..24.37% respectively, over the control. The physiological attributes of chickpea as grain N and protein content were statistically increased by 7.6% over the control, with application of T2 at moisture level 1 (55 % FC) and by 10.12(grain N) with T4 at moisture  level 3 (95 % FC). NOTE: Please check the calculation, only in N grain for T4 moisture level 3, not for protein.

Response: Needful changes have been incorporated according to suggestions of worthy reviewer at line no 512-515.

Comment 50: Line 509-512 “Also in the present study, the calculated proline content of chickpea leaves showed an increase at moisture level 1 (55 % of FC) by inoculation with T2-consortium having ACCD activity, and IAA production.” Delete EPS, since it was not tested.

Response: Needful changes have been incorporated according to suggestions of worthy reviewer at line no 518-520.

Comment 51: Line 513 ” Statistically higher grain yield (35.49 %) over that of untreated chickpea plants was obtained with T7 treatment (Mesorhizobium ciceri RZ-22 + Enterobacter Cloacae RP-08 + Providencia vermicola RS-15) at moisture level 2 (75% of FCand with T2 (Mesorhizobium ciceri RZ-11 + Bacillus subtilis  RP-01+ Bacillus mojavensis RS-14) (50% higher grain yield) at  moisture level 1 (55% of FC).”

Response: Needful changes have been incorporated according to suggestions of worthy reviewer at line no 521-525.

Comment 52: Line 517 “T4 (Mesorhizobium ciceri RZ-11 + Enterobacter Cloacae RP-08 + Providencia vermicola RS-15) had 27.53% better economic yield at moisture level 3 (95% of FC) as compared to uninoculated chickpea plants.”

Response: Needful changes have been incorporated according to suggestions of worthy reviewer at line no 526-527.

Comment 53: Line 519 “Hence, microbial combinations…”

 Response: Needful changes have been incorporated according to suggestions of worthy reviewer at line no 528.

In Conclusions:

Comment 54: Line 525 “..desert soil is a valid idea”. Delete the word “quite”.

Response: Needful changes have been incorporated according to suggestions of worthy reviewer at line no 533.

Comment 55: Line 526 “Here, series of experiments revealed growth promotion as well as substantial nodulation in chickpea that enhanced its grain yield under drought stress.”

Response: Needful changes have been incorporated according to suggestions of worthy reviewer at line no 534-535.

Comment 56:Line 528 “..under drought spell in Thal desert, a main chickpea producing area in..”

Response: Needful changes have been incorporated according to suggestions of worthy reviewer at line no 537.